# Sex-specific multi-level 3D genome dynamics in the mouse brain

Devin Rocks [1,3], Mamta Shukla [2,3], Laila Ouldibbat[1], Silvia C. Finnemann [1], Achyuth Kalluchi[2],
M. Jordan Rowley [2✉] & Marija Kundakovic [1✉]

The female mammalian brain exhibits sex hormone-driven plasticity during the reproductive period. Recent evidence implicates chromatin dynamics in gene regulation underlying this plasticity. However, whether ovarian hormones impact higher-order chromatin organization in post-mitotic neurons in vivo is unknown. Here, we mapped the 3D genome of ventral hippocampal neurons across the oestrous cycle and by sex in mice. In females, we find cycle-driven dynamism in 3D chromatin organization, including in oestrogen response elements-enriched X chromosome compartments, autosomal CTCF loops, and enhancer-promoter interactions. With rising oestrogen levels, the female 3D genome becomes more similar to the male 3D genome. Cyclical enhancer-promoter interactions are partially associated with gene expression and enriched for brain disorder-relevant genes and pathways. Our study reveals unique 3D genome dynamics in the female brain relevant to female-specific gene regulation, neuroplasticity, and disease risk.

[1] Department of Biological Sciences, Fordham University, Bronx, NY, USA. [2] Department of Genetics, Cell Biology and Anatomy, University of Nebraska Medical Center, Omaha, NE, USA. [3] These authors contributed equally: Devin Rocks, Mamta Shukla. ✉email: jordan.rowley@unmc.edu; mkundakovic@fordham.edu

Sex differences in brain physiology and disease result from the interplay of sex hormones and sex chromosome-linked genes[1]. Brain sexual differentiation is established during perinatal development and the process continues across peripubertal and adult life. From puberty to menopause, cyclical fluctuations in ovarian hormones represent a female-unique experience, and while necessary for reproductive function, they are associated with substantial behavioural and brain plasticity[2], and with increased female risk for certain brain disorders such as anxiety and depression[3].

Ovarian hormones, oestrogen and progesterone, have receptors widely distributed in the brain and exert potent neuromodulatory effects[4,5]. In rodents, the oestrous cycle is characterised by variation in dendritic spine density in the hippocampus, largely driven by fluctuating oestrogen levels[6,7]. Other aspects of hippocampal physiology, such as long-term potentiation[8] and dentate gyrus neurogenesis[9], also show cyclic patterns and are enhanced during the high-oestrogenic phase of the cycle. Recent findings in humans indicate that the structure and functional connectivity in the human female brain are comparably dynamic and vary with the menstrual cycle[10–13], closely paralleling oestrogen levels[10]. However, little is still known about molecular mechanisms underlying the sex hormone-induced, dynamic nature of the female brain.

We recently showed that chromatin accessibility, a major mechanism controlling gene expression, varies with the oestrous cycle and sex in the ventral hippocampus, a brain region essential for emotion regulation in mice[14]. We linked these sex-specific chromatin dynamics to changes in neuronal gene expression, neural plasticity, and anxiety-related behaviour[14]. However, whether sex hormones are able to dynamically change the higher-order chromatin organization in post-mitotic neurons of the brain remains unknown. Three-dimensional (3D) genome organization allows interactions of genes with their distant cis-regulatory elements, through chromatin looping and compartmentalization, and is thought to play a major role in transcriptional regulation[15–17]. Within the brain, 3D genome remodelling has only recently been implicated in neuronal differentiation[18] and function[19,20], neuronal activity-dependent gene regulation[21,22], and memory formation[23–26], but whether there are sex differences and sex hormone-mediated influences on this regulation remains unknown.

To address this question, we profiled 3D genome organization in adult ventral hippocampal (vHIP) neurons across the oestrous cycle and by sex using an unbiased chromatin conformation capture (Hi-C) method combined with DNA fluorescence in situ hybridisation (FISH) for candidate loci. We integrated 3D genome data with chromatin accessibility (ATAC-seq) and gene expression (RNA-seq) data on the same biological samples. In addition to sex differences, our study shows dramatic multi-level changes in 3D genome organization across the oestrous cycle, which are partially associated with chromatin accessibility and gene expression changes and enriched for brain disorder-relevant genes and pathways. By oestrogen replacement in ovariectomized (OVX) female mice, we confirm that the 3D genome of vHIP neurons is highly responsive to sex hormone changes. Our study reveals unique 3D genome dynamics in the female brain that has the potential to contribute to both brain and behavioural plasticity and female-specific risks for brain disorders.

## Results

**Oestrous cycle study design.** For this study, we performed the Hi-C method on vHIP neurons isolated from 11-week-old male and female mice (Fig. 1a). To explore the effect of the oestrous cycle on 3D genome dynamics in the female brain, we tracked the cycle comprehensively over three consecutive cycles[14] and included two female groups in the following oestrous cycle stages: proestrus (high oestrogen-low progesterone) and early dioestrus (low oestrogen-high progesterone)[14], reminiscent of the human follicular and luteal phase, respectively (Fig. 1a). From all three groups, vHIP tissue was first cross-linked to 'fix' protein to DNA and preserve 3D interactions, nuclei were isolated, and neuronal (NeuN+) nuclei were purified using fluorescence-activated nuclei sorting (FANS)[27]. The Hi-C assay proceeded with DNA digestion by restriction enzymes, filling the digested ends and labelling them with biotin, and ligation (Fig. 1a). Finally, the proximally-ligated DNA was fragmented, and the biotinylated fragments were enriched and used for Hi-C library preparation (Fig. 1a).

We performed bioinformatics analysis on Hi-C libraries focusing on three different levels of 3D chromatin organization within each chromosome: chromosome compartments, CTCF loops and loop domains, and enhancer–promoter (E–P) interactions (Fig. 1b). Hi-C experiments were performed in triplicates which clearly clustered by group (Fig. 1c). Considering the high correlation between the replicates in each group, we pooled the data by group for all subsequent analyses resulting in 0.53–0.70 billion sequenced reads per group with 0.36–0.48 billion useable reads after alignment and quality filtering (Supplementary Data 1). To explore the role of 3D genome organization in the regulation of gene expression in vHIP neurons by sex and oestrous cycle, the Hi-C data were integrated with our chromatin accessibility (ATAC-seq) and gene expression (RNA-seq) data, which were also generated in triplicates[14].

**Ventral hippocampal neurons display known features of 3D genome organization.** Across all samples and groups, we found that the majority (72–74%) of Hi-C interactions in vHIP neurons are intra-chromosomal, with only a limited degree of interactions found between chromosomes (26–28%, Supplementary Data 1), which is expected and consistent with the existence of distinct chromosome territories in these neurons[16]. Using the dioestrus group as a representative example, we then explored the features of 3D genome organization including chromosomal compartments, CTCF loops, and E–P interactions[16] in our Hi-C data (Fig. 1c–l, Supplementary Fig. 1). Within each chromosome, the Hi-C contact maps showed a recognisable plaid pattern of interactions, indicating chromatin segregation into two major compartments, A (active) and B (inactive) compartments[28] (Fig. 1b, d), called at 25 kb resolution. These compartments are defined by the eigenvector or first component of a principal component analysis, with positive values defining the A compartment and negative values indicating the B compartment (Fig. 1d)[28]. When Hi-C maps were overlapped with our RNA-seq data, as expected, the A compartment correlated with transcriptionally active genes while the B compartment was generally associated with inactive genes[28] (Fig. 1d, Supplementary Fig. 1a).

We next examined high-intensity chromatin interactions commonly referred to as CTCF loops (Fig. 1b) which are visually apparent as strong punctate signals in Hi-C maps[16] (Fig. 1e). Using Significant Interaction Peak caller (SIP)[29], we were able to call a total of 9721 punctate loops in vHIP neurons at 10 kb resolution (Fig. 1f, g). Across 14,794 detected loop anchors (Fig. 1f), we confirmed that the top enriched motif was the CTCF motif (Supplementary Fig. 1b) with the majority in a convergent orientation (Supplementary Fig. 1c), which is a common feature of CTCF loops[30]. These CTCF loops create domains of interactions (Supplementary Fig. 1d) as described in other cell types[29,30]. Moreover, when overlapped with ATAC-seq data, the vast majority (82%) of the called loops corresponded with both ATAC-seq peaks and the CTCF motif, while few of them

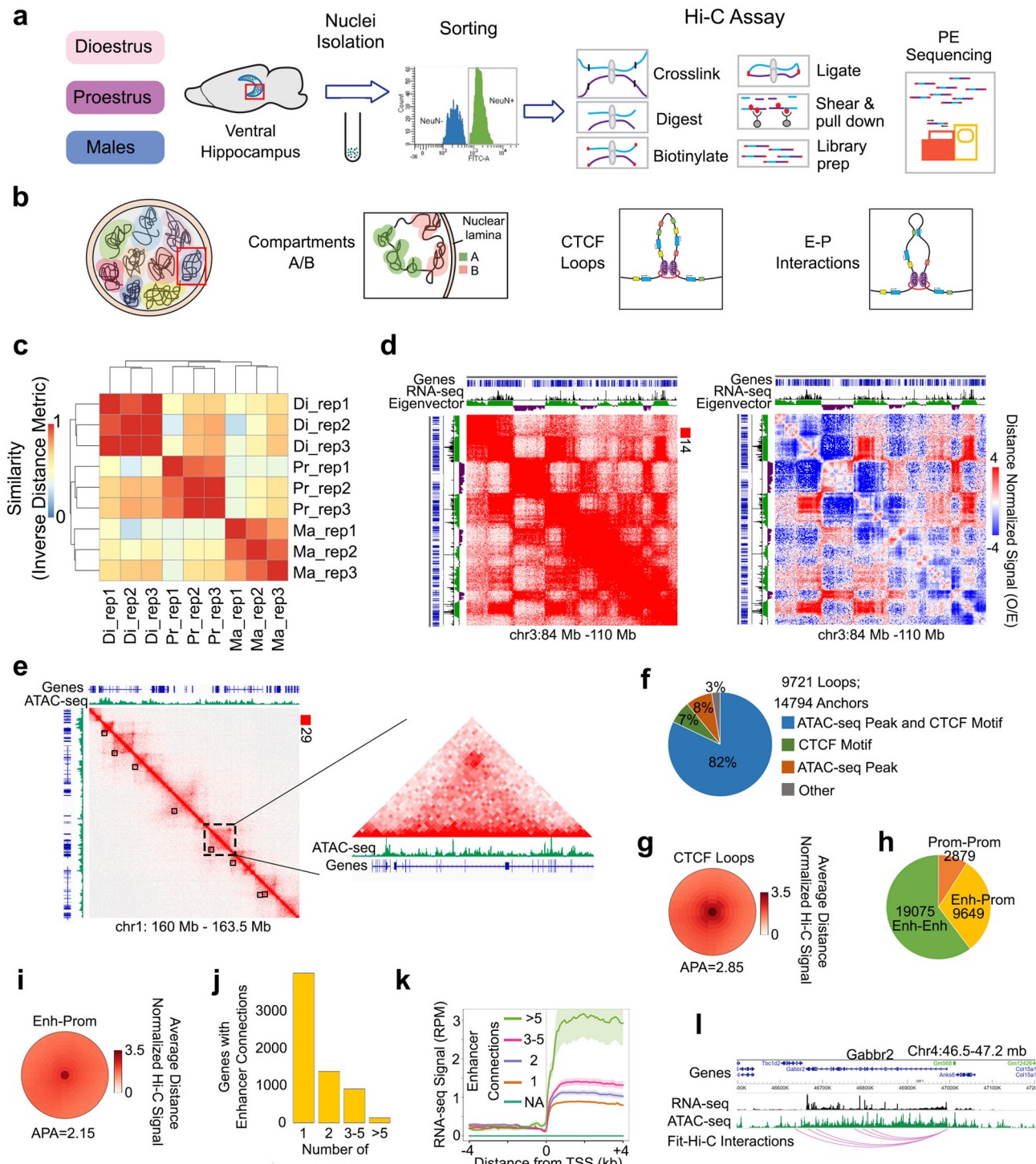

**Fig. 1 3D genome organization of vHIP neurons across the oestrous cycle and sex.** The Hi-C assay was performed on sorted neuronal (NeuN+) nuclei isolated from the ventral hippocampus of dioestrus, proestrus, and male mice (**a**) to study different levels of 3D chromatin organization within each chromosome (**b**). Biological replicates ($n = 3$/group) clustered per group (**c**) and the pooled dioestrus group was selected as a representative to explore 3D genome organization. The Hi-C contact matrix (left) and distance-normalised matrix (right) are presented with the RNA-seq data overlaid onto the eigenvector signal; the plaid pattern and corresponding gene expression profile indicate the presence of two compartments, active and inactive, called at 25 kb resolution (**d**). The high-intensity punctate signals represent CTCF loops, identified at 10 kb resolution (black squares) and shown with overlapping ATAC-seq signal (**e**), with the majority of loop anchors containing both an ATAC-seq peak and CTCF motif (**f**); the average distance-normalised loop signal is shown (**g**). As a third component of chromatin organization, enhancer–promoter (E–P) interactions (**h**) exhibit a weaker distance-normalised Hi-C signal (**i**) and the number of these interactions varies across genes (**j**) and correlate with gene expression (shaded area represents standard error) (**k**). *Gabbr2* is represented as an example of a gene with multiple E–P connections and high expression levels (**l**). IGV tracks show merged RNA-seq and ATAC-seq data with the corresponding E–P interactions derived from three biological replicates of the dioestrus group.

corresponded with either CTCF motif only (7%), ATAC-seq peak only (8%), or neither criteria (3%, Fig. 1f).

Finally, we used Fit-Hi-C2[31] to call E–P interactions in vHIP neurons with 10 kb resolution. For this purpose, we used non-promoter ATAC-seq peaks, ≥10 kb from transcription start sites (TSSs), as a proxy for enhancers. To justify our choice, we show that published H3K4me1 and H3K27ac data derived from sorted hippocampal neurons[32] identify these peaks as loci with the marks of active enhancers (Supplementary Fig. 1e), consistent with our previous findings in cortical neurons[33]. The E–P interaction analysis resulted in a total of 19,075 enhancer–enhancer (E–E) interactions, 9649 E–P interactions, and 2879 promoter–promoter (P–P) interactions (Fig. 1h). On average, the intensity of the E–P, E–E, and P–P interactions (APA score = 2.15, 2.08, and 2.26; Fig. 1i, Supplementary Fig. 1f) was weaker than that of the CTCF loops (APA = 2.85; Fig. 1g). We also found that the number of enhancer connections varied among genes (Fig. 1j) and was directly correlated with gene expression level in vHIP neurons (Fig. 1k). In particular, the highest RNA-seq signal was detected in genes having numerous (>5) enhancer interactions (Fig. 1k, Supplementary Data 2), and these multi-enhancer genes showed an enrichment for pathways important for neuronal function and hormone signalling (Supplementary Fig. 1g). In fact, by exploring the data from the mouse neural differentiation study by Bonev et al.[18] we found that the expression of our multi-enhancer genes is likely to emerge with neuronal lineage specification (Supplementary Fig. 1h–j). These genes were found to be enriched for repressive H3K27me3 histone marks in embryonic stem (ES) cells while this Polycomb-mediated repression appears to be lost in neuronal precursor cells (NPCs) and cortical neurons[18] (CNs, Supplementary Fig. 1h, i). We also found that the multi-enhancer interactions that we identified in our vHIP neurons may be emerging during the acquisition of neuronal fate; these interactions are stronger in NPCs than in ES cells and are further strengthened upon neuronal differentiation (Supplementary Fig. 1j). In general, when compared across tissues, our identified multi-enhancer genes show highest expression in the brain including in the cerebral cortex and hippocampal formation[34] (Supplementary Fig. 1k). As an example, a highly expressed, brain-specific gene Gabbr2, encoding a subunit of the GABA-B receptor, interacts with multiple (6) putative enhancers located in accessible chromatin regions up to 360 kb downstream of the Gabbr2 promoter (Fig. 1l, Supplementary Fig. 1l).

In summary, we show that the Hi-C data from sorted vHIP neurons display known features of 3D chromatin organization, including chromosomal compartments, CTCF loops, and E–P interactions, which are strongly associated with chromatin accessibility states and gene expression in these cells.

**Compartmental organization of the X chromosome is sex- and oestrous cycle dependent**. We next compared the compartmental organization of vHIP neurons across the three groups—dioestrus, proestrus, and males (Fig. 2, Supplementary Fig. 2). We found no large difference in the compartmental structure of autosomes in either the male to female (Male vs. Die) or within-female (Pro vs. Die) comparisons (Supplementary Fig. 2a–d). Chromosome 14 showed some small differences in compartment signal (Supplementary Fig. 2a, e) which is consistent with reports that chromosome 14 shares epigenetic features with sex chromosomes[35]. We found that the sex chromosome-related genes previously found on chromosome 14[35] have larger differences in compartment structure than other genes on this chromosome (Supplementary Fig. 2f), suggesting that sex-specific autosomal genes may have nuanced sex-specific compartmental

structures. We then found that the compartmental organization of the X chromosome varies with both sex and the oestrous cycle (Fig. 2a, b, Supplementary Fig. 2a, b, g). The sex difference in the X chromosome was expected considering that the eigenvector signal for females is a mixed signal from the active (Xa) and inactive (Xi) X chromosome, while the same signal in males originates from a single (and active) X chromosome (Fig. 2a). Although the difference in X chromosome eigenvector in the male to female (Male vs. Die) comparison was more profound than that in the within female (Pro vs. Die) comparison (Fig. 2a), differences were large in both comparisons in terms of both the overall signal intensity (Fig. 2b) and the number of bins being affected (Supplementary Fig. 2g). In the proestrus to dioestrus comparison, 1081 bins or 15.9% (27 Mb) of the X chromosome showed changes in compartment signal (Supplementary Fig. 2g). These differences in compartmental signal were further associated with differences in chromatin accessibility (Supplementary Fig. 2h).

Interestingly, when comparing Pro vs. Die, we also found that the differences in compartmental signal were often going in the same direction as Male vs. Die differences (Fig. 2c), suggesting that the X chromosome compartmental signal in the high-oestrogenic proestrus female group is more similar to that of males, when compared to the low-oestrogenic dioestrus group. Furthermore, we found that the X chromosome in males and proestrus females showed more inter-chromosomal interactions than the X chromosome in the dioestrus group (Fig. 2d). All described similarities between proestrus and males were specific to the X chromosome and were not found in autosomes (Fig. 2b, Supplementary Fig. 2i, j), implying that proestrus may be associated with a higher volume or partial decondensation of the Xi chromosome, making the X chromosome compartmental signal in proestrus more similar to that in males.

To confirm the observed compartmental change in the X chromosome across the oestrous cycle, we performed four-colour FISH. Based on our Hi-C data, we designed a control DNA probe (p4) located in chromosome 1, and three DNA probes located in the X chromosome: the probes p2 and p3 were in the compartments A and B, respectively, in all three groups; and, the probe p1 was found to be in the compartment B in males while it showed a 'switch' between compartments B (Pro) and A (Die) during the oestrous cycle in females (Fig. 2e, f). We quantified our FISH data by subtracting the physical distance between probes p1 and p3 from the distance between the probes p1 and p2 (Fig. 2f, g). We found that p1 and p2 are more physically separated in the proestrus and male samples than in the dioestrus samples (Fig. 2f, g), confirming the altered compartmental profiles derived from the Hi-C method (Fig. 2e). Interestingly, these oestrous cycle-driven changes may be brain region-specific, as the same DNA-FISH assay in the visual cortex did not show a significant difference between proestrus and dioestrus, despite confirming the sex difference in the physical distance among the tested probes (Supplementary Fig. 3).

We further explored whether gene expression changes may be associated with the observed sex- and oestrous cycle-dependent X chromosome compartmental differences in vHIP neurons. Naturally, we first looked into X chromosome escapee genes which are able to escape the inactivation of the Xi chromosome and are, thus, more highly expressed in females than in males[36]. Taking the top 5 escapee genes from the Die-Male comparison (Fig. 2h, Supplementary Fig. 4a), we indeed found that these genes are always located in the A compartment in dioestrus (Supplementary Fig. 4b) and show a higher eigenvector A compartment signal compared to expression matched non-escapee genes (Fig. 2i, Supplementary Fig. 4c, d). In males, these genes did not show heightened A compartment signal compared

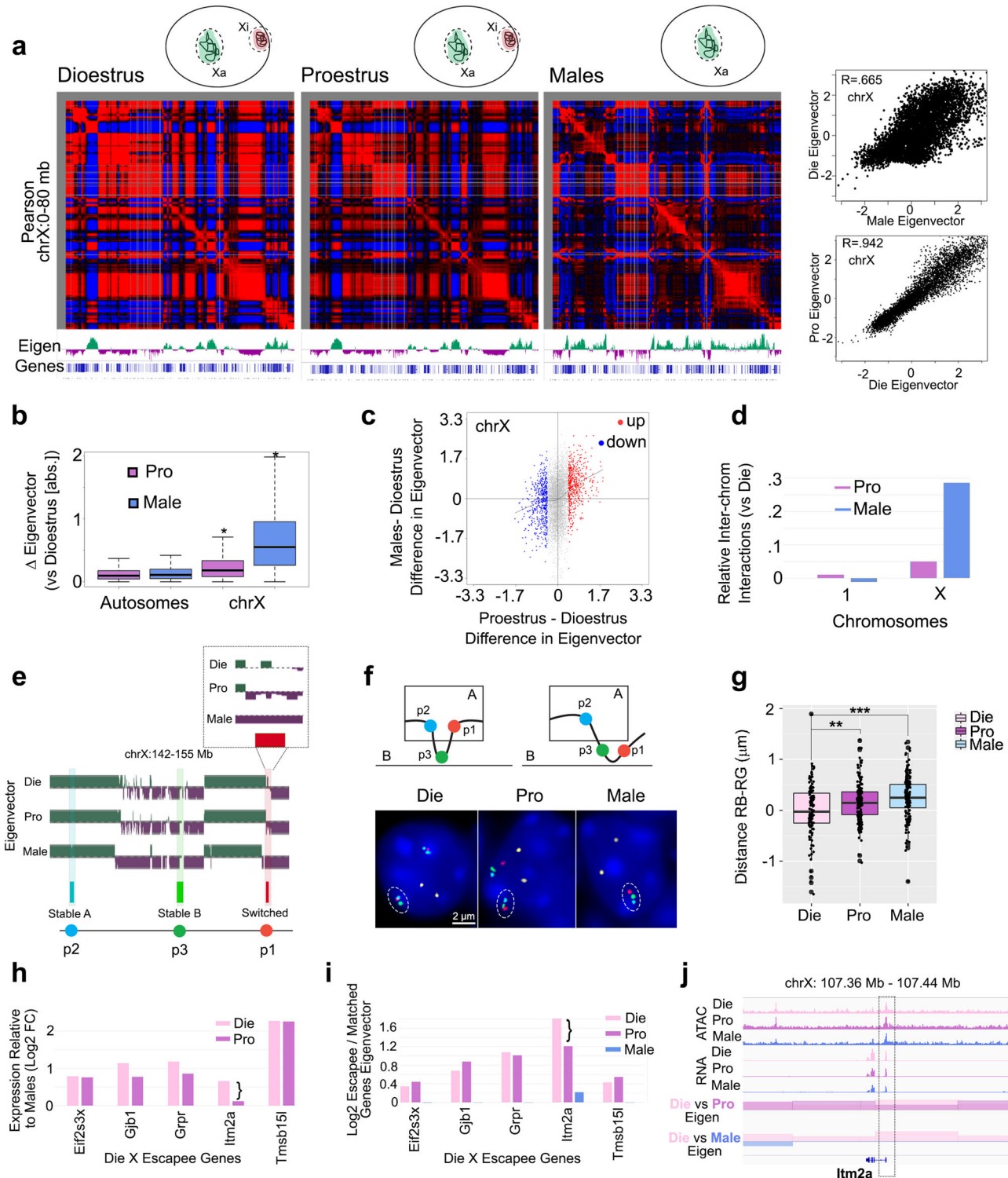

to other genes (Fig. 2i, Supplementary Fig. 4c, d). Four of these genes (*Eif2s3x*, *Gjb1*, *Grpr*, and *Tmsb15l*) also showed a higher expression and heightened eigenvector signal in proestrus females compared to males (Fig. 2h, i). However, *Itm2a* expression during proestrus was reduced compared to dioestrus; this was associated with a decreased eigenvector signal in the Pro-Die comparison making the signal of the proestrus female group, again, more similar to that of males (Fig. 2h–j). Curiously, we did not find differential ATAC-seq peaks at the promoters of escapee genes, suggesting that differences in compartmental interactions, rather

than accessibility, better explains the escape of the genes from dosage compensation (Supplementary Fig. 4e).

We also checked the *Htr2c* gene which is located nearby the FISH probes that we designed (Fig. 2e) and is another gene that is differentially expressed in the Die-Male but not in the Pro-Male comparison[14] (Supplementary Fig. 4f). Importantly, this gene, too, showed a change in the signal between dioestrus and proestrus groups, with the proestrus signal being more similar to males, particularly in the area adjacent to the TSS and the end of the gene (Supplementary Fig. 4f).

**Fig. 2 Compartmental organization of the X chromosome in vHIP neurons across the oestrous cycle and sex.** The correlation matrix and the corresponding eigenvector signal (Eigen) show compartmental profiles along an 80 Mb-segment of the chromosome X in dioestrus, proestrus, and male groups (**a**). Females have an active (Xa) and an inactive (Xi) X chromosome while males have only one active (Xa) chromosome (shown in inset), which is consistent with the largely different eigenvector signal in the Die-Male comparison ($R = 0.665$) and more similar profile in the Pro-Die comparison ($R = 0.942$, panels on the right). The eigenvector signal is different in the X chromosome ($n = 6267$ bins) but not in autosomes ($n = 102,529$ bins) in both Pro-Die and Male-Die comparisons; Wilcoxon rank-sum test; *$P < 0.05$ (**b**). Proestrus and male signals go in the same direction when compared to the dioestrus signal (**c**). X chromosome, but not chromosome 1, shows more interactions with other chromosomes in males and proestrus compared to dioestrus (**d**). The compartmental profile of a smaller 13-Mb segment of X chromosome is depicted in Die, Pro, and Male, and the location of the designed FISH probes (p1–p3) is shown (**e**) and the expected compartmental change of the region surrounding probe 1, a switch from compartment A to B in Die vs. Pro, was confirmed with FISH (representative images are shown; $n = 3$ animals/group; scale bar: 2 μm; note: yellow signal is a negative control on chromosome 1) (**f**). FISH data were calculated as the RB-RG distance by subtracting the distance between the centre of the red and green signals (RG) from that of the red and blue signals (RB). The analysis was restricted to X chromosomes positive for all three probe signals ($n = 116$ Die; $n = 154$ Pro; $n = 163$ Male); one-way ANOVA with post hoc Tukey; **$P < 0.01$ ($P = 0.00775$); ***$P < 0.001$ ($P = 0.0000129$). Source data are provided as a Source Data file (**g**). The expression (**h**) and relative eigenvector signal (**i**) of the top five X escapee genes was shown in dioestrus and proestrus females compared to males. *Itm2a* gene is shown as an example of X escapee that changes expression and compartmental signal across the oestrous cycle (**j**). IGV tracks show merged ATAC-seq and RNA-seq data with the corresponding eigenvector signal comparisons for dioestrus (Die), proestrus (Pro), and males (Male) derived from 3 biological replicates for each group. Box plots (box, 1st–3rd quartile; horizontal line, median; whiskers, 1.5× IQR). Die, pink; Pro, purple; Male, blue.

Finally, to explore which upstream regulators may be driving the observed compartmental differences, we performed the motif analysis of the genomic areas showing differential compartmental signal between the sexes and across the oestrous cycle. Interestingly, we found the sex-determining region Y gene (Sry) and Egr2 binding sites to be enriched in the Die-Male comparison (Supplementary Fig. 4g). In the Die-Pro comparison, we found binding sites for the transcription factor Pou3f2 as well as for the response elements for oestrogen receptor alpha (ERα, ESR1 motif) and oestrogen-related receptor 2 (ERR2, Supplementary Fig. 4h), consistent with hormonal regulation of the higher-order chromatin organization in vHIP neurons across the oestrous cycle in females.

In summary, we found a significant compartmental change in oestrogen response elements-enriched X chromosome genomic areas during the oestrous cycle, with proestrus females showing a compartmental profile more similar to males, which was partially associated with gene expression.

**Neuronal CTCF loops vary with sex and the oestrous cycle.** We next examined the effect of sex and the oestrous cycle stage on the CTCF loops in vHIP neurons (Figs. 3, 4). For the between-sex comparison, we explored comparing either the dioestrus group to males or the mixed-female group (merged dioestrus and proestrus) to males. We found an increased ability (1.65 times) to call sex-specific loops when comparing dioestrus to males, as opposed to comparing mixed females to males (Fig. 3a, Supplementary Fig. 5a), indicating that separating females by the oestrous cycle stage helps identify sex differences in chromatin looping. Out of 9721 called loops, we found 260 loops to be stronger in dioestrus than in males (Die-specific) and 366 loops to be stronger in males than in dioestrus females (Male-specific, Fig. 3b). Loop signal in proestrus was between that of dioestrus and males for both categories of loops, but also with many loops specific to proestrus (Supplementary Fig. 5b) which we examine later (Fig. 4). Importantly, unlike compartmental differences, only 6 of these sex-specific loops were on X chromosomes and all others were on autosomes (Fig. 3c). Intriguingly, CTCF motifs at differential loops were frequently in non-convergent orientations (Supplementary Fig. 5c). Additional motif analysis found oestrogen response elements (the ESR1/2 motif) to be enriched at the dioestrus-specific loop anchors while the ZFP770 motif was enriched at the male-specific loop anchors (Fig. 3d). Finally, we explored whether gene expression differences were associated with sex-specific loops. We closely examined a dioestrus-specific loop encompassing 8 genes and 3.5 Mb at chromosome 1

(Fig. 3e). Interestingly, we found little difference in the expression of the genes interior to the loop, but this may be explained by the high number of strong interior loops that display no evident differences between dieoestrus and males (Fig. 3e). There were more profound differences in the expression of two genes near the CTCF loop anchors, but outside of the loop, including Erbb4 (Fig. 3e), the gene encoding the neuregulin receptor. However, while specific loops were associated with transcriptional differences, in general, there was little correlation between differential loops and gene expression (Supplementary Fig. 5d). To understand this further, we first performed genome-wide Aggregate TAD Analysis (ATA) as well as plotted insulation scores at their borders, but found no evident differences in these domain signals or insulation (Supplementary Fig. 5e). To more specifically investigate how these differential CTCF loops may impact insulated loop domains, we examined insulation scores and performed aggregate domain analysis (ADA) at differential loops. Curiously, changes in the loops did not result in obvious differences in insulation scores or loop domains (Supplementary Fig. 5f, g). This is consistent with these differential loops representing non-convergent loops which are less important for interaction domains[29,37], but acting more precisely similar to E–P interactions. This finding is also consistent with recent evidence suggesting that CTCF loops are likely only a small component of interaction domains and gene expression control[30,38]. This may be the reason why we see no genome-wide changes to the expression of genes that are interior to differential loops (Supplementary Fig. 5d) but do detect changes in expression of *Errb4* and *Mreg*, which are next to differential loop anchors but located outside the loop (Fig. 3e).

We further explored differential CTCF loops in the within-female comparison and found roughly the same number of differential loops in the Die-Pro comparison (Fig. 4a) as we found in the Die-Male comparison (Fig. 3b). Located primarily on autosomes (Fig. 4b), 370 loops were stronger in dioestrus than in proestrus (Die-specific) and 264 loops were stronger in proestrus than in dioestrus (Pro-specific, Fig. 4a). Of the Pro-specific loops, 84 were called differential in the Die-Male comparison (Male > Die), leaving 180 loops that were proestrus only (Supplementary Fig. 5a). However, when we compared their signal intensity compared to dioestrus, the loop signal changes in proestrus were largely in the same direction as the loop signal changes in males, simply to a different degree (Fig. 4c). In addition, the motif analysis found oestrogen response elements (the ESR1/2 motif) to be enriched in both dioestrus-specific and proestrus-specific loops (Fig. 4d). Using confocal microscopy, we confirmed that the

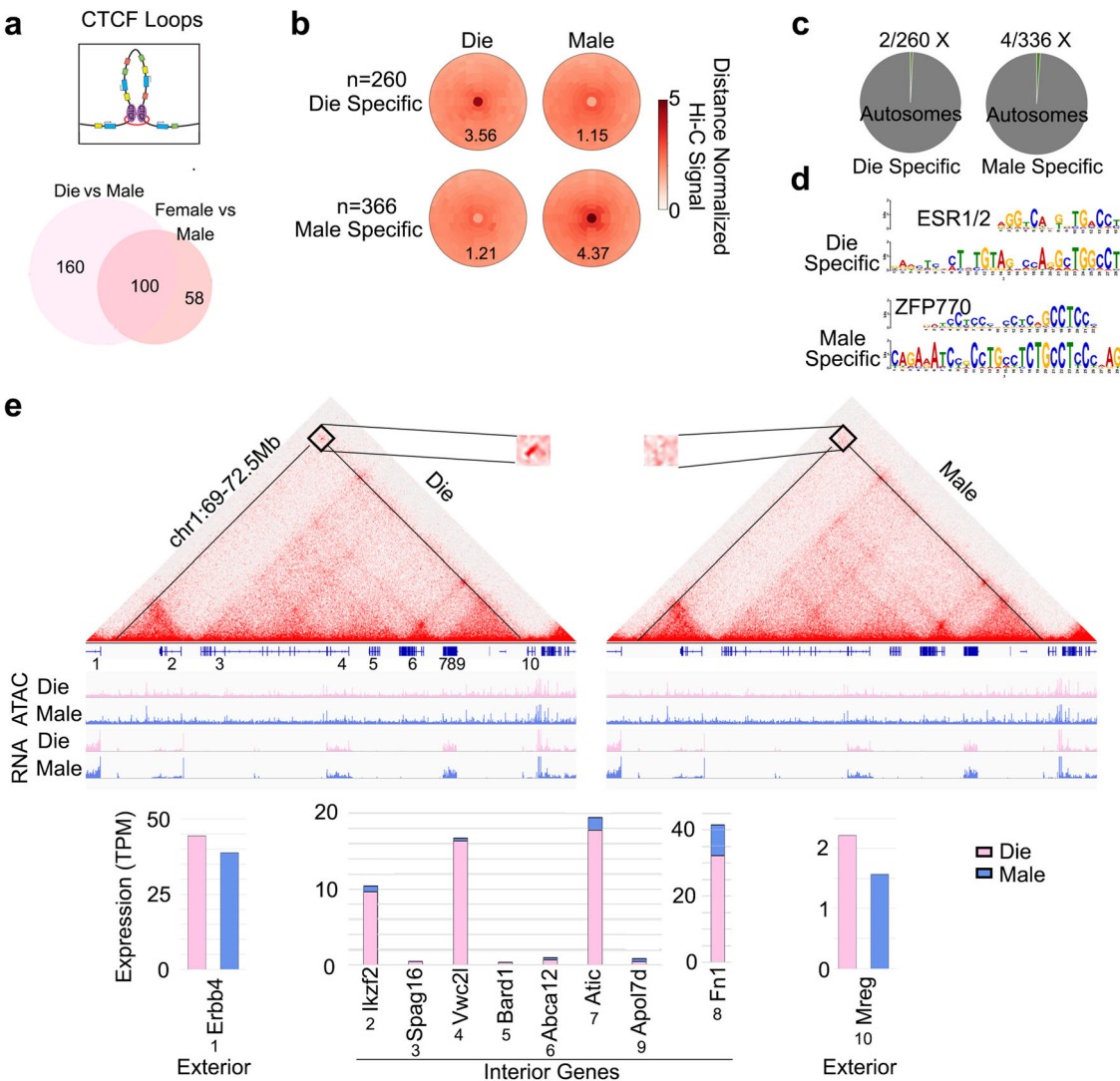

**Fig. 3 Sex-specific CTCF loops in vHIP neurons.** More sex-specific loops can be called in the Die-Male than in the Mixed Female–Male comparison (**a**). Both Die- and Male-specific loops (**b**) are primarily located on autosomes (**c**). Die-specific loops are enriched for the ERα (ESR1) motif (E value 6.9e−33) while Male-specific loops are enriched for ZFP770 motif (E value 5.0e−59) (**d**). Presented is a 3.5 MB loop encompassing eight interior genes and flanked by two exterior genes (**e**). IGV tracks show merged ATAC-seq and RNA-seq data with the corresponding loops for dioestrus (Die) and males (Male) derived from three biological replicates for each group. Lower panel shows gene expression for each exterior and interior gene. Die, pink; Male, blue.

oestrogen receptor alpha (ERα) localises to the nucleus of vHIP neurons (Fig. 4e, Supplementary Fig. 5h), consistent with its role as a transcription factor and possible regulator of chromatin loops.

Similar to what we found for sex-specific loops, there was, in general, little correlation between differential Pro-Die loops, insulation scores, loop domains, and gene expression (Supplementary Fig. 5i-k). The lack of change to insulation or domain structure is likely due to the same reasons we saw lack of changes in these features in the Die-Male comparison as mentioned above. We also note that many of these loops are "nested" in that they share loop anchors with other loops that are unchanged between samples which likely stabilises insulation at these anchors (Fig. 4f, g, Supplementary Fig. 5l, m). However, we were able to identify specific loops associated with transcriptional differences. An interesting example is a sex-specific and oestrous cycle-dependent loop involving *Adcyap1*, an important stress- and oestrogen-sensitive gene implicated in anxiety[39,40] (Fig. 4f). This 2 Mb loop connects *Adcyap1* and an upstream region of *Mettl4*, is stronger in proestrus and males than in dioestrus, and corresponds to

differential *Adcyap1* expression among the three groups (Fig. 4f, Supplementary Data 3). Notably, another CTCF loop that connects *Adcyap1* and *Mettl4* is present in all groups and may be more important for regulating *Mettl4* expression which shows no expression difference across the groups (Fig. 4f). It is worth noting that the observed differential loop also appears in the mixed-female (Die+Pro) to male comparison (Supplementary Fig. 5m), further highlighting that the sex-specific dynamism that we see with proestrus becoming more similar to the male *Adcyap1* gene looping (and the associated gene expression change) is only discoverable if the assay resolution is increased by monitoring the oestrous cycle stage (Fig. 3a). Another example includes a loop surrounding the *Thbs2* gene (Fig. 4g). Similar to *Adcyap1*, the *Thbs2* gene is also an anxiety-related gene[41] that has an oestrogen response element in the promoter and can be regulated by ERα[42]. While *Thbs2* is more highly expressed in dioestrus in comparison to both proestrus and males, we found a proestrus-specific loop (Fig. 4g, Supplementary Data 3) that is likely to be regulated in a sex-specific way, by varying sex hormone levels in females.

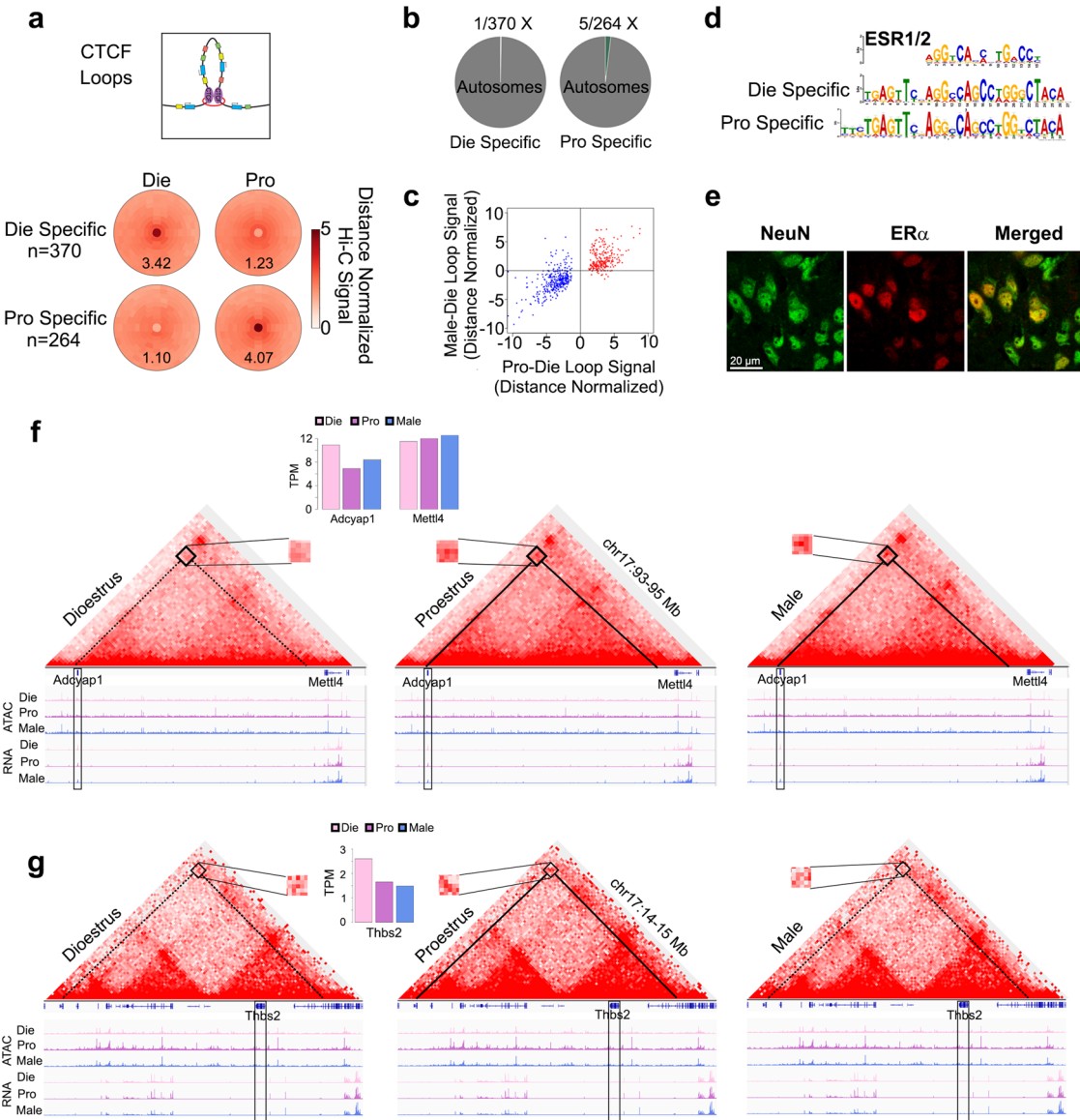

**Fig. 4 Oestrous cycle-dependent CTCF loops in vHIP neurons.** Oestrous cycle stage-dependent, Die- and Pro-specific loops (**a**) are primarily located on autosomes (**b**); Pro-Die loop signal goes in the same direction as Male-Die signal (**c**). Die- and Pro-specific loops are enriched for the ERα (ESR1) motif (E values 1.1e−62 and 2.5e−58, respectively) (**d**). ERα localises to NeuN+ nuclei in the ventral hippocampus (**e**, representative proestrus sample is shown; $n = 3$ animals/group; scale bar: 20 μm). Two example genes *Adcyap1* (**f**) and *Thbs2* (**g**) are presented. Differential loops are depicted with either a solid line (strong loop) or dashed line (weak loop). IGV tracks show merged ATAC-seq and RNA-seq data with the corresponding loops for dioestrus (Die), proestrus (Pro), and males (Male) derived from 3 biological replicates for each group. Insets show gene expression for *Adcyap1, Mettl4* (neighbouring gene), and *Thbs2*. Die, pink; Pro, purple; Male, blue.

In conclusion, we found sex- and oestrous cycle-specific CTCF loops in vHIP neurons that are partially associated with gene expression changes and are likely to be regulated by oestrogen receptors in females.

**Neuronal enhancer–promoter interactions vary with sex and oestrous cycle.** Finally, we explored differential E–P interactions and, strikingly, found around 2000 differential E–P interactions in both Die-Male and Die-Pro comparisons (Fig. 5a). Again, when compared to dioestrus, the proestrus E–P signal was largely going in the same direction as the male E–P signal (Fig. 5b). We found oestrogen response elements (the ESR1 motif) to be enriched in dioestrus-specific and proestrus-specific EP interactions (Fig. 5c), and these elements also appeared as top motifs in Die-to-Male differential EP interactions (Supplementary Fig. 6a). In

general, EP differences did not show high correlation with differential gene expression neither in Die-Pro (Supplementary Fig. 6b) nor in Die-Male (Supplementary Fig. 6c) comparisons, indicating that these differential E–P interactions are not sufficient for changing gene expression, likely due to the ability of promoters to be regulated by multiple enhancers. However, looking specifically at genes that do show differential expression, we found that around 10% of genes whose expression varies with the oestrous cycle[14] showed concomitant, cycle-driven differential E–P interactions (Supplementary Data 3). For instance, the *Pou3f2* gene, an important psychiatric risk-related gene encoding a brain-specific transcription factor, showed clear oestrous cycle-dependent (Fig. 5d) and sex-specific (Supplementary Fig. 6d) E–P interaction profiles which were associated with differential gene expression. We further looked into several additional genes with

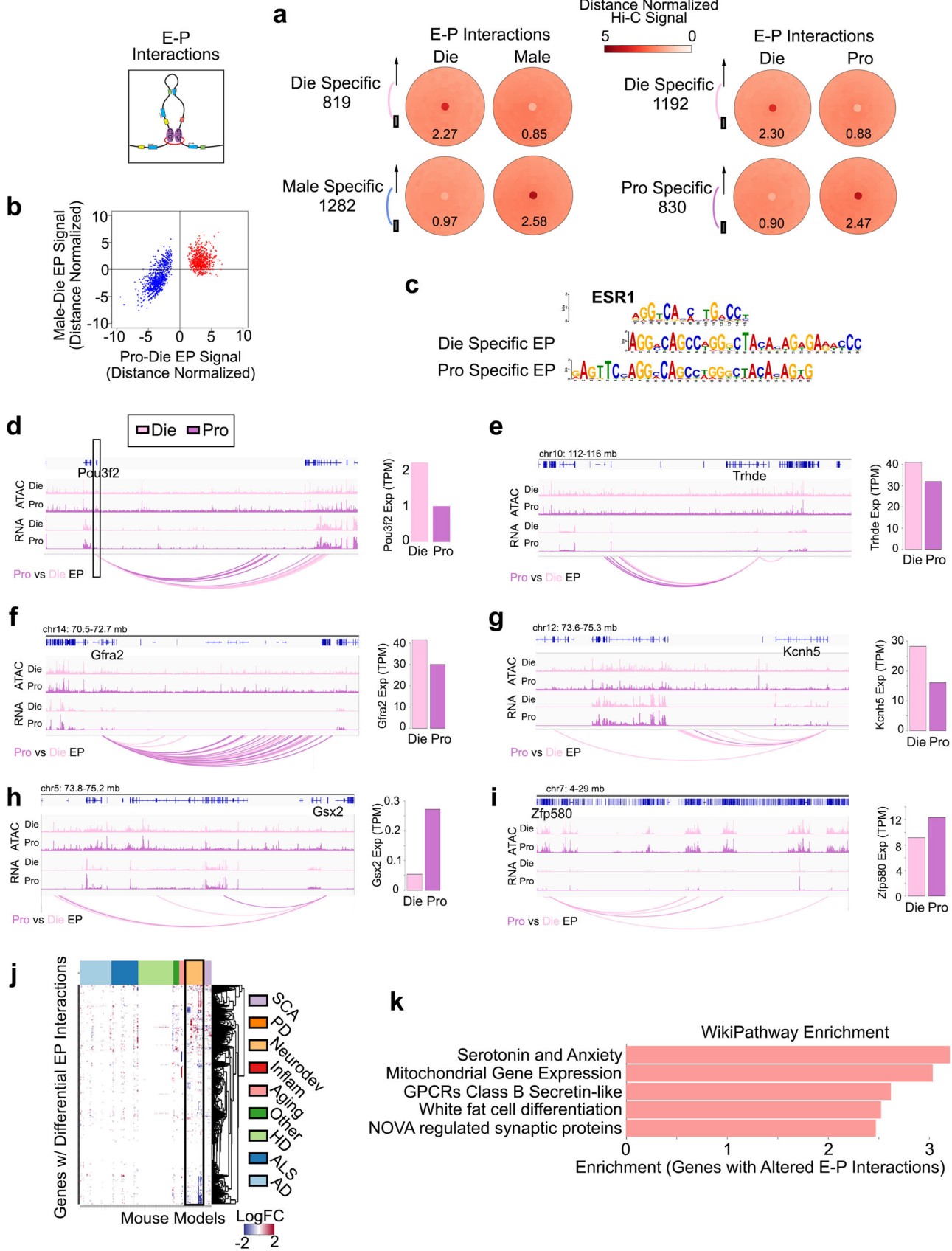

**Fig. 5 Dynamics of E–P interactions in vHIP neurons across the oestrous cycle.** The number of differential E–P interactions across sex and the oestrous cycle is comparable (**a**) and the E–P signal goes in the same direction in the Pro-Die and Male-Die comparisons (**b**). Across the oestrous cycle, in differential Die- and Pro-specific E–P interactions, there is an enrichment of binding sites for ERα (ESR1) motif (E values of 1.2e−85 and 3.2e−48, respectively; **c**). Example genes showing differential E–P interaction profiles and differential gene expression across the oestrous cycle include Pou3f2 (**d**), Trhde (**e**), Gfra2 (**f**), Kcnh5 (**g**), Gsx2 (**h**), and Zfp580 (**i**). IGV tracks show merged ATAC-seq and RNA-seq data with the corresponding E–P interactions for dioestrus (Die) and proestrus (Pro) derived from three biological replicates for each group. Bar graphs (on the right) show gene expression for each gene. Cyclical E–P interactions are enriched for genes altered in mouse models relevant to neurological disorders (**j**) and several gene pathways including 'Serotonin and Anxiety' as the top pathway (**k**).

cyclical expression and important neuronal function including *Trhde* (encoding a thyrotropin-releasing hormone degrading enzyme), *Gfra2* (encoding a GDNF family receptor), *Kcnh5* (encoding a potassium voltage-gated channel), *Gsx2* (encoding the GS homeobox 2), and *Zfp580* (encoding the Zing finger protein 580, Fig. 5e–i). While some of these genes showed numerous E–P interactions (e.g. *Gfra2*, Fig. 5f) and others showed few of them (e.g. *Gsx2*, Fig. 5h), the number of which, again, appeared to be associated with the gene expression level, differential interactions were typically associated with the distant genomic regions and correlated with transcriptional differences in an oestrous cycle-dependent (Fig. 5e–i) and sex-specific manner (Supplementary Fig. 7).

Finally, we examined which genes and pathways are enriched within oestrous cycle- and sex-specific E–P interactions, irrespective of gene expression profiles (Fig. 5j, k). We first used a mouse RNA-seq portal that includes mouse transcriptomic datasets related to multiple neurological disorders and aging models[43]. Interestingly, many of the genes with differential E–P interactions across the oestrous cycle, while not showing gene expression changes in our physiological mouse model, exhibit aberrant expression in mouse models of neurological diseases (Fig. 5j). In addition, the top pathway shown to be enriched in the dioestrus-proestrus comparison was the serotonin and anxiety pathway, further suggesting that the genes relevant to serotonergic function and anxiety-related behaviour overwhelmingly show 3D genome organizational changes across the oestrous cycle (Fig. 5k, Supplementary Fig. 8a). This pathway was specific to within-female comparison, as we found a more generic G-protein-coupled receptors pathway to be the top enriched pathway in the between-sex (Die-Male) comparison (Supplementary Fig. 8b).

In summary, we found thousands of oestrous cycle-dependent and sex-specific E–P interactions that are partially associated with gene expression differences and are enriched for oestrogen response elements and brain disease-relevant genes and pathways.

**Oestrogen replacement in ovariectomized (OVX) female mice partially recapitulates proestrus-associated 3D genome changes.** Since we found the enrichment of oestrogen response elements in the oestrous cycle-dependent compartments, loops, and E–P interactions (Supplementary Fig. 4h, Figs. 4d, 5c), we further explored the role of oestrogen in female-specific 3D genome dynamics in vHIP neurons (Figs. 6, 7). These experiments were performed in adult OVX female mice in order to remove the effect of endogenous ovarian hormones, which are typically completely depleted 3 weeks following the ovariectomy[44]. We asked the question whether short-term oestradiol treatment could recreate 3D genome changes observed during the high-oestrogenic, proestrus phase of the cycle. The oestrogen treatment regimen (4 h following s.c. injection of 5 μg oestradiol benzoate EB) was selected based on: a) the oestrous cycle-driven oestrogen's temporal dynamics; and b) earlier studies showing oestrogen-induced ERα genomic binding and transcription in the mouse brain within this time frame[45]. Importantly, we also

confirmed that this oestrogen regimen induces changes in behaviour in cycling, low-oestrogenic (dioestrus) animals (Fig. 6a), practically mimicking the physiological, proestrus-associated decrease in anxiety indices that we previously described[14]. We confirmed that vehicle-treated OVX mice had low or undetectable levels of serum oestradiol which was in stark contrast with EB-treated OVX mice (Fig. 6b). Then we performed the 3D genome analysis on both groups with the procedure identical to that performed with cycling females and males, including vHIP dissection, neuronal (NeuN+) nuclei isolation, the Hi-C assay, and bioinformatics analysis (Fig. 6b). We included five biological replicates per group which, after pooling, resulted in roughly 1.45 billion sequenced reads per group and ~0.88 billion useable reads per group after alignment and quality filtering (Supplementary Data 4).

We first called chromosome compartments and, strikingly, observed changes in compartmental organization in oestrogen-treated OVX mice (Fig. 6c–e) which were comparable to the changes we found across the natural cycle (Fig. 2, Supplementary Figs. 2, 4). Oestradiol-induced fairly large changes in compartments along the X chromosome (Fig. 6c) and no changes in autosomes with an exception of chromosome 14 (Fig. 6c), which was previously shown to have sexually dimorphic compartment organization (Supplementary Fig. 2a, e), likely through features shared with the X chromosome[35]. Overall, we found 230 X chromosome compartments to overlap between the Die-Pro comparison (total of 1081 differential compartments) and the EB-Vehicle comparison (total of 686 differential compartments, Fig. 6d). Within the overlapped compartments, there was an enrichment for the ERR2 motif (Fig. 6d), which was previously found to be enriched in the differential X compartments between dioestrus and proestrus groups (Supplementary Fig. 4h). Moreover, the same probe region (p1) that exhibited a compartmental switch (A to B) in the Die-Pro comparison (Fig. 2e), and was confirmed using the FISH assay (Fig. 2f, g), showed a comparable (A-B) switch after EB treatment of OVX animals (Fig. 6e). Together, these data strongly indicate that the X chromosome compartmental changes in vHIP neurons during the oestrous cycle are largely driven by the oestradiol level changes.

By analyzing CTCF loops, we found that our 4 h EB treatment alters 819 loops in vHIP neurons of OVX animals (Fig. 7a), the extent of which is similar to the number of differential loops found across the oestrous cycle (Fig. 4a). Since we wanted to determine whether acute oestrogen treatment can mimic proestrus-associated 3D changes, we focused on proestrus-specific loops in the Die-Pro comparison (Fig. 7b) and found that 23.9% (63/264) of these loops are also changed by EB treatment in OVX females, with the majority being in the expected direction (becoming stronger in response to EB, Fig. 7c).

We performed a similar analysis of E–P interactions, where we first focused on proestrus-specific E–P interactions in the Die-Pro comparison (Fig. 7d). When we compared genes showing differential E–P interactions in the Die-Pro comparison (2496 genes) with those showing differential interactions in the EB-vehicle comparison (1770 genes), we found a significant overlap

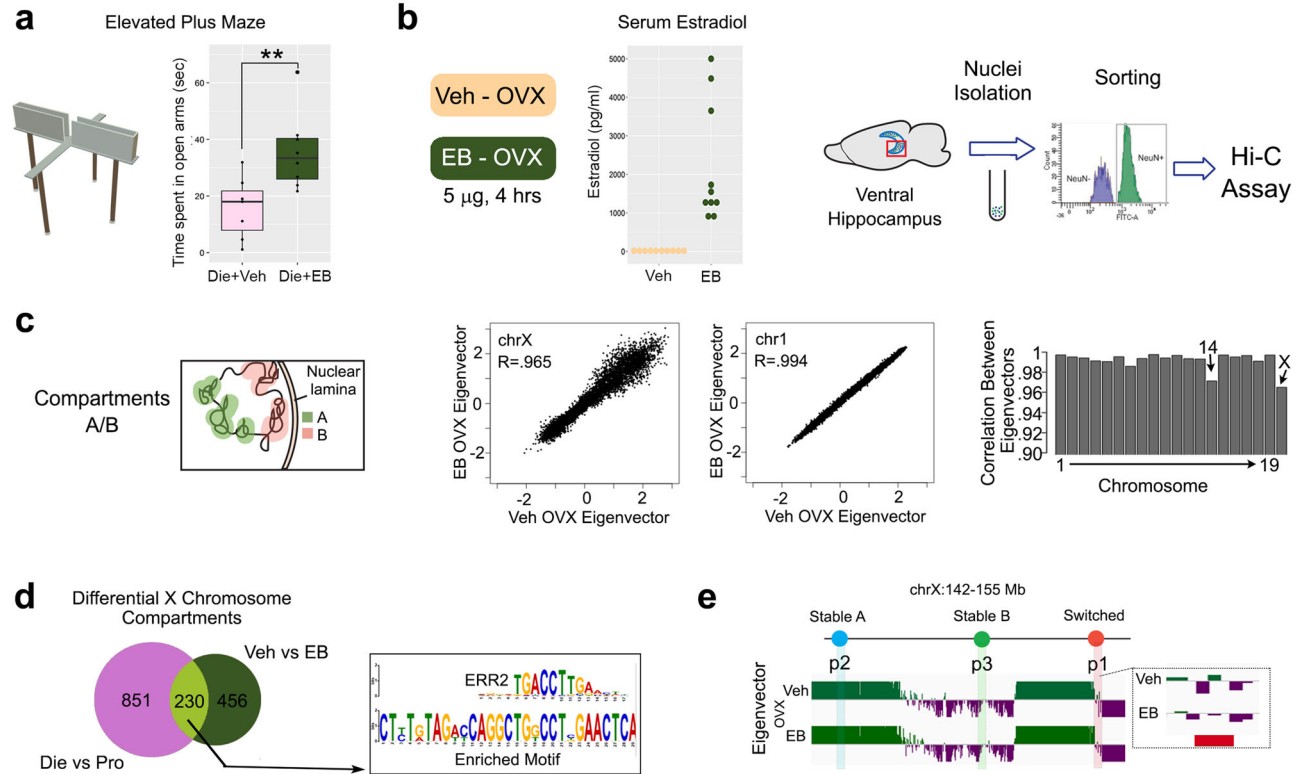

**Fig. 6 Oestradiol induces proestrus-like X chromosome compartmental changes in vHIP neurons of OVX animals.** An acute oestradiol benzoate (EB) treatment regimen was shown to induce (**a**) proestrus-like behavioural change in low-oestrogenic cycling (Die) mice. Box plots (box, 1st–3rd quartile; horizontal line, median; whiskers, 1.5× IQR); paired-sample two-sided *T* test; **$P < 0.01$ ($P = 0.00795$); $n = 7$ Die + Veh; $n = 8$ Die + EB. Source data are provided as a Source Data file. The same regimen was (**b**) administered in OVX mice (EB-OVX) and confirmed to reach high serum oestradiol levels compared to vehicle-treated OVX mice (Veh-OVX). Source data are provided as a Source Data file. Hi-C assay was then performed on vHIP neuronal nuclei isolated from both mouse groups ($n = 5$ biological replicates/group). The analysis of compartments (**c**) indicated the most profound change in the X chromosome while, among autosomes, only chromosome 14 showed notable changes. The X chromosome compartments that overlap between Die-Pro and EB-Vehicle comparisons are enriched for the ERR2 motif (E value 1.3e−8, **d**) and include the p1 probe region used for the FISH analysis (**e**). Die, pink; EB, green; OVX-Vehicle, yellow.

of 192 genes (Fig. 7e, Supplementary Data 5). Interestingly, when we performed motif analysis of enhancers associated with these overlapped genes, we found the enrichment of the binding site for Egr1 (Fig. 7e), an oestrogen-responsive transcription factor that we previously implicated in chromatin accessibility regulation across the cycle[14]. Among the overlapped genes, we highlight *Gfra2* that shows a large number of E–P interactions, which vary across the oestrous cycle (Fig. 5f) and this is, in part, recreated by EB treatment in OVX animals (Fig. 7f).

Finally, we note that the overlap between changes in 3D genome organization across the cycle (Figs. 2, 4, 5) and those induced by EB treatment in OVX animals (Figs. 6, 7), while obvious, is far from complete and includes 15–25% of overlap across the three levels of organization (Figs. 6d, 7c, e). To explore a possible reason for the limited data overlap, we down-sampled OVX Hi-C data (Supplementary Data 4) to ensure equal number of sequencing reads in all samples and then compared the overall Hi-C signal in cycling animals with that in OVX animals (Supplementary Fig. 9a). Importantly, the PCA analysis showed that the overall Hi-C signal is much more similar between dioestrus and proestrus groups compared to that of the OVX samples (both vehicle and EB-treated, Supplementary Fig. 9a), indicating that neuronal 3D genome organization undergoes large changes following ovariectomy-induced depletion of endogenous sex hormones. Interestingly, the changed responsiveness of 3D genome to oestrogen following ovariectomy is also mirrored in the changed behavioural response. Indeed, while our 4 h EB

treatment increased the time spent in the open arms of the elevated plus maze (a reduced anxiety index) in low-oestrogenic cycling female mice (Fig. 6a), this EB regimen was not sufficient to change anxiety-like behaviour in OVX mice (Supplementary Fig. 9b). The OVX mice did have a behavioural response to oestrogen, which included overall lower activity levels following EB treatment (Supplementary Fig. 9b), which was not seen in the cycling animals (Supplementary Fig. 9c). This data further indicated that OVX mice undergo brain adaptations that change their response to oestrogen compared to the animals undergoing the regular reproductive cycles.

In summary, we show that a short-term oestradiol exposure of OVX animals can, in part, recreate proestrus-specific 3D genome changes, confirming the role of oestrogen fluctuation in the dynamic 3D genome reorganization across the oestrous cycle. Our findings also highlight the importance of studying the physiological oestrous cycle in order to understand the dynamics of the 3D genome in the female brain.

## Discussion

3D genome organization allows orderly interactions of physically-distant parts of the genome and is thought to play a critical role in gene regulation and cellular function across organs and disease states[15,17]. 3D genome remodelling has only recently been implicated in brain development[18] and function[19,20], neuronal activity-dependent gene regulation[21,22], and memory formation[23–26]. However, previous 3D genome studies of the

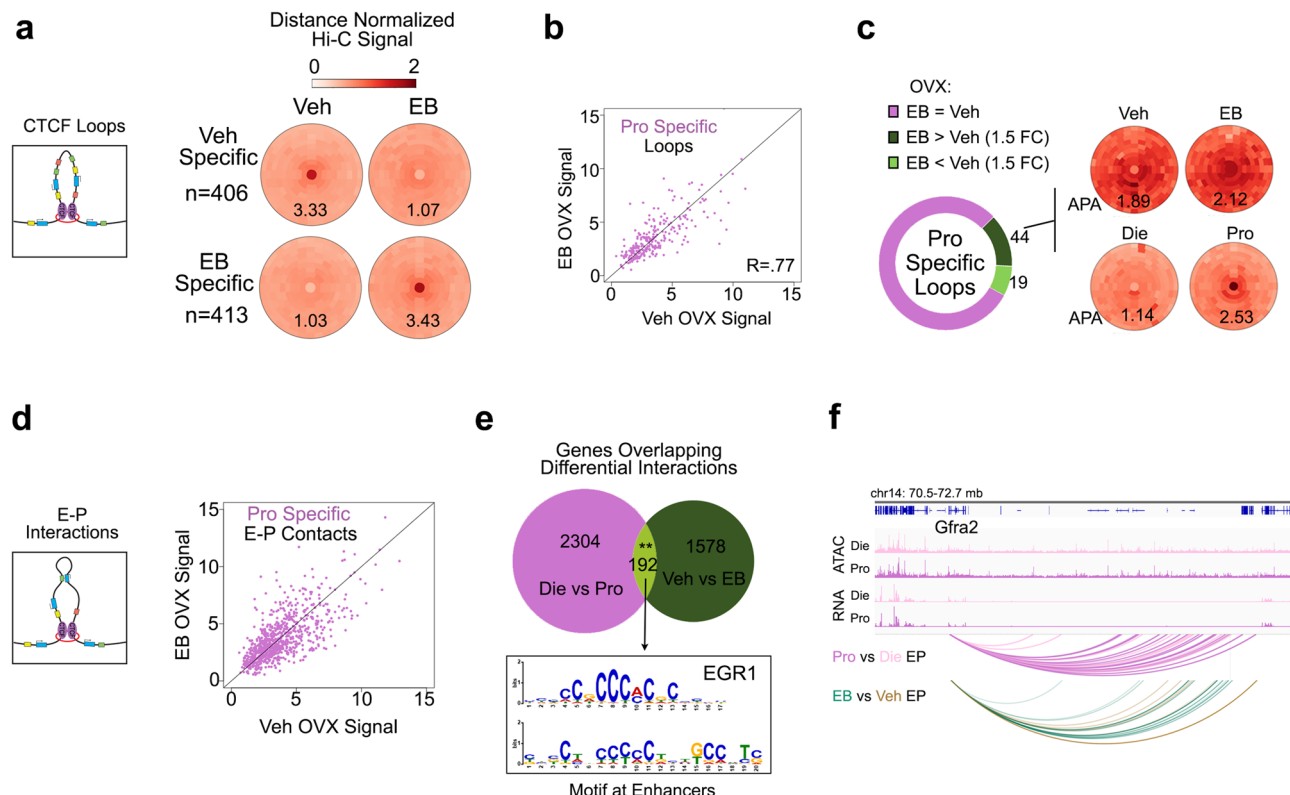

**Fig. 7 Proestrus-like, oestradiol-induced changes in CTCF loops and E–P interactions in vHIP neurons of OVX animals.** Acute oestradiol benzoate (EB) treatment induces changes in CTCF loops in OVX animals (**a**), which partially overlap with proestrus-specific loops (**b**), and the majority of the overlapped loops are stronger in EB-treated animals (**c**). EB also changes E–P interactions that partially overlap with proestrus-specific E–Ps (**d**). The gene overlap between differential E–P interactions across the cycle (Die vs. Pro) and after EB treatment in OVX mice (Veh vs. EB) is small but significant and shows an enrichment for Egr1 motifs at the enhancers (E value 6.0e−4, **e**). Example gene *Gfra2* showing differential E–P interaction profiles across the oestrous cycle (Pro vs. Die; pooled three biological replicates/group) and after EB treatment (EB vs. Veh; pooled five biological replicates/group) (**f**). IGV tracks show merged ATAC-seq and RNA-seq data for dioestrus (Die) and proestrus (Pro) derived from three biological replicates for each group.

brain were either focused on the male brain or did not explore sex differences. Here we provide evidence that 3D chromatin structure in the brain differs between males and females and undergoes dynamic remodelling during the female oestrous cycle.

Among different levels of chromatin organization, the compartmental organization of chromosomes is thought to be the most stable one in fully differentiated cells such as post-mitotic neurons[46]. Here we found minimal sex- or oestrous cycle-driven difference in autosomes, and an expected sex difference in the X chromosome. The largest autosomal compartment differences were on chromosome 14 which has genes that share regulatory features with the sex chromosomes[35], indicating that 3D chromosomal organization reflects sexually dimorphic features. There was also a surprisingly high degree of X chromosome compartmental dynamics across the oestrous cycle. A body of evidence shows that the Xi chromosome in females has a condensed 3D structure and is associated with the nuclear lamina, while the Xa chromosome is more centrally-located in the nucleus, open, and active[47]. Using Hi-C and FISH as two complimentary approaches to study 3D genome organization[48], we show that, with naturally rising oestrogen levels in female mice, the X chromosome changes compartmental organization, acquires an interaction profile more similar to males, and is likely to undergo a decondensation event and increase in volume during proestrus.

While we observe associated gene expression changes, particularly in X escapee genes, the significance and possible implications of hormonally-driven X chromosome dynamics may go well beyond the regulation of X chromosome-linked genes. For instance, it was proposed that the Xi chromosome may be

affecting autosomal gene expression by sequestering heterochromatin factors in the cell[49], thus chromatin changes in Xi may affect gene regulation more globally. It is also worth noting that X chromosome decondensation was reported in oestrogen-dependent breast cancers[50], in which case the observed deleterious X chromosome-reactivation[51] could be an extreme case of what oestrogen does under physiological conditions. In addition, the variability of X chromosome inactivation was reported to be higher than expected[52], including spatially- and temporally-variable escapee genes, which at least in part may be due to the natural variation in hormonal levels driving gene regulation. Together, we anticipate that physiological dynamics in the X chromosome may include global changes in cellular environment affecting neuronal gene regulation and function and this warrants further investigation as it may have important implications for female health and disease. It is also worth noting that we were able to partially recreate the cycle-driven X chromosome compartmental reorganization by oestrogen replacement in female mice following ovariectomy. These data not only confirm our oestrous cycle data but also open an intriguing possibility of hormonally-mediated Xi chromosome reactivation that may be of interest for the treatments of X-linked developmental disorders such as Rett syndrome[53].

Intriguingly, the oestrous cycle- and sex-driven compartment changes on the X chromosome are not composed of altered CTCF loops, and therefore do not support the hierarchical model of chromosome organization. Instead, our results are more consistent with a model where compartments and CTCF loops are independent layers of genome organization[16]. This model has

been supported by studies where degradation of looping factors results in little to no change in compartments[30,54]. These earlier studies also showed that de novo identification of chromatin contact domains by Hi-C, sometimes referred to as topologically associating domains (TADs), often represents a mixture of CTCF loop domains and compartment domains[16,30,55,56]. The definition of 'TADs' is of interest, but is discussed elsewhere and is not the purpose of this study[56]. However, we acknowledge that, in recent years, the term 'TAD' is becoming more synonymous with CTCF loop domains. Curiously, despite differences in CTCF loops, we do not detect changes in contact domains, i.e. Genova-identified 'TADs' nor in CTCF loop domains. This is likely because cycle-specific loops tend to share one or more anchors with unchanging loops. Therefore, the oestrous cycle-driven changes to CTCF loops are likely important for priming the regulation of specific genes nearby, rather than the larger numbers of genes found inside loop domains.

In general, we observe that oestrous cycle- and sex-driven 3D genome changes at all levels of organization are associated with changes in expression of relevant genes. For instance, we see cycle- and sex-dependent compartmental and gene expression changes in the gene encoding the serotonin receptor 2c (*Htr2c*). This receptor is known to play an important role in anxiety-related behaviour[57], and these data are consistent with our previous findings that dioestrus females exhibit higher anxiety indices compared to proestrus females and males[14]. Furthermore, changes in CTCF looping around genes such as *Adcyap1* and *Thbs2* are associated with sex- and oestrous cycle-dependent gene expression differences and further provide a possible link between 3D genome changes and anxiety-related behavioural phenotypes. In particular, *Adcyap1* encodes pituitary adenylate cyclase activating polypeptide (PACAP), a critical sex-specific and oestrogen-dependent regulator of the stress response[58] which has been associated with sex-specific anxiety-related behaviour in rodents[39] and female-specific risk for PTSD in humans[40]. Moreover, we observed differential E–P interactions and changes in the expression of genes such as *Pou3f2*[59], *Trhde*[60], and *Gfra2*[61], which are associated with psychiatric disorders in humans and anxiety and depression-related behaviours in mice. While our E–P results were obtained by examining significant interactions in Hi-C data, future work using methods that are specific for E–P interactions, such as HiChIP[62] for H3K4me1 or H3K27ac, may identify additional sex specific- or sex hormone-driven E–P interactions critical for gene regulation and neuronal function. Overall, considering the importance of 3D genome organization for gene regulation and cellular function, our results provide a molecular mechanism for sex specific- and sex hormone-driven neuronal gene regulation, neural plasticity, and behavioural phenotypes.

When considering possible mechanisms that drive 3D genome changes in a sex specific- and oestrous cycle-dependent manner, it is striking that we found the consistent enrichment of oestrogen response elements across all levels of analysis including compartments, CTCF loops, and E–P interactions. Oestrogen regulates genes through either classic nuclear oestrogen signalling or through membrane-bound oestrogen receptors (ERs)[4]. Our previous findings from the ATAC-seq analysis implicated a membrane-bound ER-mediated mechanism and Egr1 transcription factor in the regulation of chromatin accessibility across the oestrous cycle[14]. However, current Hi-C data strongly imply that 3D genome organization in vHIP neurons in females is driven by nuclear ERα. We know that both oestrogen levels and ERα expression vary with the oestrous cycle[14,63] and this is likely reflected in the differential, cycle-dependent ERα binding to the genome, thus providing a likely mechanism for changing 3D chromatin organization with varying sex hormone levels. Indeed,

studies in breast cancer cell lines showed that the binding of ERs has an instructive role in chromatin looping[64] and have proposed a role for steroid receptors as genome organizers at a local and global scale[65]. Consistent with this, Achinger-Kawecka et al. found that 3D genome remodelling during the development of endocrine resistance in ER+ breast cancer cells is mediated by ERα binding[66], further linking ERα with 3D genome organization. With our oestrogen replacement experiments, we confirm the ability of oestrogen to induce 3D genome changes at all levels of organization. Unsurprisingly, consistent with other ovariectomy-induced brain adaptations[67–69], the 3D genome in OVX animals shows an altered state and differential responsiveness to oestrogen compared to the 3D genome of naturally-cycling females. Actually, these data further confirm that the higher-order chromatin organization in neurons is highly responsive to sex hormone changes in terms of both increased hormonal levels as well as hormonal withdrawal. Nevertheless, we see important overlaps between proestrus- and oestrogen replacement-induced 3D genome changes, and reveal the Egr1 motif as being enriched at the enhancers of the overlapped genes. Intriguingly, these data suggest that membrane-bound and nuclear-bound ER pathways may interact through Egr1 in chromatin and gene regulation across the oestrous cycle.

Importantly, we also see a lot of changes in 3D genome interactions that are not associated with gene expression changes in our physiological model. These 'non-functional' 3D organizational changes, therefore, may represent an 'epigenetic priming event' or could be part of cellular memory associated with cycling events. For instance, CTCF has been shown to be critical for synaptic plasticity and memory formation[70,71]. In studies of mice with disruptions in the CTCF loop organizer, it was shown that memory-relevant genes do not exhibit hippocampal gene expression changes under basal conditions but are affected in activity-dependent way[24]. Other studies also showed learning[23]- and immunological memory[72]- related 'epigenetic priming' where epigenomic changes precede changes in gene expression which require another stimulus to be expressed. This molecular priming is consistent with female-specific, reproductive-related physiology where many events across the ovarian cycle including ovarian changes, thickening of the uterus wall, and likely brain structural changes are preparatory rather than functionally relevant events. Strikingly, we also see that oestrous cycle-dependent (but not sex-specific) differential E–P interactions are enriched for neurological disorder- and serotonin and anxiety-relevant genes. Both clinical and basic studies show that the increased female risk for depression and anxiety disorders is associated with fluctuating sex hormone levels[3]. While varying sex hormones by themselves may not be sufficient to trigger the disorder, they may 'prime' the female brain for increased vulnerability that can be precipitated by stress and other risk factors. Therefore, the molecular priming effects that we see in the form of 3D chromatin organizational changes across the ovarian cycle may represent the molecular basis for female-specific vulnerability for certain brain disorders such as anxiety and depression.

In summary, our study reveals female-specific 3D genome dynamics that has both functional and priming effects on the expression of the neuronal genome and a potential to contribute to reproductive hormone-induced brain plasticity and female-specific risk for brain disorders.

## Methods

**Animals**. For experiments involving intact cycling females, male and female C57BL/6J mice from Jackson Laboratory arrived at the Fordham University Animal Facility at 7 weeks of age. For experiments involving ovariectomized (OVX) female mice, C57BL/6J females underwent ovariectomy at Jackson Laboratory at 8 weeks and arrived at our facility at 9 weeks of age. All mice were housed in same-

sex cages ($n = 4$–$5$ per cage) in a room set to 21 °C ambient temperature and 30–70% humidity, habituated for 2 weeks, and were kept on a 12:12 h light:dark cycle (lights on at 8 a.m.) with *ad libitum* access to food (Envigo 7012) and water. For oestrous cycle experiments, the oestrous cycle of female animals was tracked daily in the morning (between 9 a.m. and 11 a.m.) for 2 weeks (between 9 and 11 weeks of age) in order to establish a predictive cycling pattern for each female animal and to ensure that only females with regular cycles were included in the study. OVX animals were briefly treated with either oestradiol benzoate or vehicle at 11 weeks of age. All male and female animals were sacrificed via cervical dislocation at 11 weeks of age; brains were extracted, and bilateral ventral hippocampi were dissected on ice then flash frozen in liquid nitrogen. For histology experiments, the whole brain was dissected and preserved for cryosectioning (see *Tissue preservation for histology*). Frozen tissue was stored at −80 °C before further processing. All animal procedures were approved by the Institutional Animal Care and Use Committee at Fordham University.

**Oestrous stage determination**. The mouse oestrous cycle is typically 4–5 days in duration and contains the following four phases: proestrus, oestrus, metoestrus, and dioestrus. Oestrous cycle stage of female animals was determined using vaginal smear cytology, as previously described[14]. Briefly, smears were collected by filling a disposable transfer pipette with 100 µl of distilled water, gently placing the tip of the pipette at the vaginal opening and collecting cells via lavage. The cell-containing water was then applied to a microscope slide and allowed to dry at room temperature for 2 h. Once dried, slides were stained with 0.1% crystal violet in distilled water, washed, and then allowed to dry prior to examination with light microscopy. Oestrous cycle stage can be determined by the relative quantities of nucleated epithelial cells, cornified epithelial cells, and leucocytes[14]. Establishing the cycling pattern of each female animal allowed us to select days for tissue collection that would maximise the number of females in the proestrus and dioestrus phases. Proestrus and early dioestrus were selected as groups for molecular and histological analysis due to their hormonal profiles mimicking the follicular and luteal phases of the human menstrual cycle, respectively. We have previously confirmed that proestrus represents the high oestradiol-low progesterone phase and early dioestrus represents the low oestradiol-high progesterone phase[14]. Phase predictions were confirmed after sacrificing by collecting and analysing post-mortem vaginal smears.

**Oestradiol injections**. For oestradiol-injection experiments, ovariectomized or dioestrus female mice were injected subcutaneously with 5 µg oestradiol benzoate (EB) or vehicle (100 µl of 1% ethanol in corn oil). Injections were administered between 9 a.m. and 11 a.m. Four hours after injections, animals were either sacrificed for Hi-C analysis or underwent behavioural testing (see 'Elevated Plus Maze').

**Oestradiol concentration measurements**. During tissue collection from oestradiol- or vehicle-injected ovariectomized females used for the Hi-C assay, trunk blood was collected. Serum was collected from blood by allowing the blood to clot for 1–2 h at room temperature, followed by centrifugation ($1500 \times g$ for 20 min at 4 °C). The supernatant serum was removed and stored at −80 °C prior to the analysis. Quantification of serum oestradiol levels using high-performance Liquid Chromatography with Tandem Mass Spectometry (HPLC/MS–MS) was performed by OpAns, LLC (Durham, NC). Briefly, oestradiol was prepared as an individual stock solution at 1 mg/mL in acetonitrile (ACN) and dimethyl sulfoxide (DMSO) (50:50), which was used to generate calibrators (ranging from 5 pg/ml to 5000 pg/ml) and quality control samples. Stable isotopic labelled internal standard was added to calibration standards, quality control, and matrix samples. The analyte and its internal standard were extracted by liquid-liquid extraction and evaporated to dryness then reconstituted and derivatized with 2-fluoro-1-methylpyridinium p-toluensulfonate prior to analysis in an LC/MS–MS system. Oestradiol derivative in the samples was separated using reversed-phase liquid chromatography with detection of the eluate by tandem mass spectrometry. Chromatographic separations were performed using a 1290 series HPLC system (Agilent Technologies, Santa Clara, CA, USA) and mass spectrometric analysis was performed using an Agilent 6495 Series Triple Quadrupole tandem mass spectrometer. HPLC/MS–MS data were acquired and processed using the proprietary software application MassHunter Workstation Data Acquisition for Triple Quad B.07.01/Build 7.1.7112.0 (Agilent Technologies, Inc.).

**Nuclei isolation and fluorescence-activated nuclei sorting (FANS) for Hi-C assay**. Purification of neuronal nuclei was performed as described previously[27] with some modifications to include formaldehyde cross-linking. Briefly, for the oestrous cycle experiment, 6 animals per each group - proestrus, dioestrus, and males (total $n = 18$)—were used for Hi-C analysis. From each animal, we used bilaterally dissected ventral hippocampi and pooled brain tissue from two animals for each biological replicate ($n = 3$ replicates/group). Nuclei preparation and sorting were performed in three batches with each batch having equal group distribution ($n = 1$ for proestrus, dioestrus and males). For oestradiol replacement experiments, we used 10 animals from each group—OVX + vehicle and OVX + oestradiol (total $n = 20$)—pooled into 10 biological replicates ($n = 5$ replicates/

group) which were sorted in four batches ($n = 2$–$3$ replicates/batch). Brain tissue was dissociated in lysis buffer using a douncer, then incubated in 1% formaldehyde for 10 min, followed by quenching with 200 mM glycine for 5 min. After washings and filtration through 70 µm strainer (Sigma), nuclei preparation continued as described previously[27]. Total nuclei were extracted using sucrose gradient centrifugation. The nuclei pellet was resuspended in DBPS and incubated, for 45 min, with the mouse monoclonal antibody against neuronal nuclear marker NeuN conjugated to AlexaFluor 488 (1:1000, MAB377X; MilliporeSigma). Before sorting, we added DAPI (1:1000) to the incubation mixture and filtered all samples through a 35-µm cell strainer. FANS was performed on a FACSAria instrument (BD Sciences) and data were collected and analysed using BD FACSDiva v8.0.1 software at the Albert Einstein College of Medicine Flow Cytometry Core Facility. In addition to a sample containing NeuN-AlexaFluor 488 and DAPI stain, three controls were used to set up the gates for sorting: DAPI only; IgG1 isotype control-AlexaFluor 488 (1:1000, Mouse monoclonal IgG1-k, FCMAB310A4; MilliporeSigma) and DAPI; and NeuN-AlexaFluor 488 only. We set up the protocol to remove debris, ensure single nuclear sorting (using DAPI), and select the NeuN+ (neuronal) and NeuN- (non-neuronal) nuclei populations (Supplementary Fig. 10a). For each biological replicate, we collected 200,000 NeuN+ (neuronal) nuclei in BSA-precoated tubes filled with 200 µL of DPBS. The purity of sorted single nuclei was confirmed using fluorescence microscopy (Supplementary Fig. 10b).

**Hi-C assay**. After FANS, 200,000 neuronal nuclei were pelleted in a centrifuge at $2850 \times g$ for 10 min at 4 °C, supernatant was removed, and the pellet was frozen in liquid nitrogen and stored at −80 °C. Cross-linked, sorted nuclei were then used for the Hi-C assay which was performed using the Arima Hi-C kit (Arima Genomics), according to the manufacturer's instructions. Briefly, the cross-linked chromatin was digested using a restriction enzyme cocktail; digested ends were filled in and labelled with biotin, followed by the ligation of spatially proximal digested ends. The proximally-ligated DNA was then purified using SPRIselect DNA purification beads (Beckman Coulter) and the first quality control checkpoint was performed to ensure that: (a) the sufficient fraction (>15%) of proximally-ligated DNA was labelled with biotin; and (b) the output of the Hi-C assay was of the expected size of 2.5–8 kb. The proximally-ligated DNA was then fragmented using a Covaris S2 instrument, targeting the DNA fragment size of 400 bp, which was confirmed using the Agilent Bioanalyzer instrument. This was followed by DNA size selection (200–600 bp) using SPRIselect beads. The size-selected biotinylated fragments were enriched with the Enrichment Beads (provided in the Arima Hi-C kit) and used for Hi-C library preparation. As recommended for low-input Hi-C protocols, the Hi-C library was prepared using the Swift Biosciences Accel 2S Plus DNA Library Kit and Indexing kit reagents, using a modified library preparation protocol in which DNA remains bound to the Enrichment Beads. Following end repair and adapter ligation and before PCR amplification, the second quality control was performed using the KAPA Library Quantification kit (Roche), to determine DNA recovery and number of PCR cycles needed for each library. The library amplification step included 6-7 PCR cycles and was performed using the KAPA Library Amplification kit (Roche). The final Hi-C library was purified using SPRIselect beads and the quality and size of the library (~500 bp) were confirmed using the Agilent Bioanalyzer. The libraries were quantified by Qubit HS DNA kit (Thermofisher Scientific) and the KAPA Library Quantification kit prior to sequencing. 150 bp, paired-end sequencing was performed on a Nova-Seq 6000 instrument at the New York Genome Centre. For the oestrus cycle experiments, a library pool containing equal amounts of all nine libraries ($n = 3$ for proestrus, dioestrus, and males) was prepared and loaded on two S1 lanes, yielding close to 200 million reads per library (Supplementary Data 1). For oestradiol replacement experiments, a library pool containing equal amounts of the ten libraries ($n = 5$/group) was prepared and loaded on one S4 lane, yielding, on average, close to 300 million reads per library (Supplementary Data 4).

**Hi-C data analysis**. Sequenced reads were mapped using BWA[73] to the mouse genome build mm10, with a quality filter ≥10, and removal of PCR duplicates. Individual replicates were assessed by comparing the distance-normalised signal as derived by the formula (observed + 1)/(expected + 1), after which the similarity metric and clustering was derived using DESeq2[74]'s sample-to-sample distance function. Replicates were combined with random sampling of reads to have equal levels between samples, and then visualised with Juicebox. Compartments were identified by calculating the eigenvector on the Pearson correlation matrix in 25 kb bins. Differential compartments were identified if the eigenvector differences were higher than 0.42 which corresponded to the differences between quartiles; in essence differential compartments were those that shifted from one quartile to another. CTCF loops were identified using SIP[29] at 10 kb resolution using the following parameters: -norm KR –g 1 –mat 2000 -fdr .01. Differential CTCF loops were called by first merging loops from each sample to a master loop list keeping only one entry if loops were called within 25 kb of each other. Then to call strong differential loops, we required a distance-normalised signal >2 and a fold change between samples ≥2.5. E–P interactions were identified by FitHiC2[31] at 10 kb resolution keeping interactions with a q vaule < 0.05 and removing any that could be considered (within 20 kb) a CTCF loop instead. We then categorised interactions based on overlap with promoters and non-promoter ATAC-seq peaks as a proxy for enhancers. Differential E–P interactions were identified for FitHiC2

interactions with a distance-normalised signal >2 and a fold change between samples ≥2.5. Comparison to Hi-C and H3K27me3 in ES, NPC, and CN was done by reprocessing data from GEO accession GSE96107[75] and from ENCODE accession ENCSR059MBO. Metaplots, Aggregate Peak Analysis (APA), and Aggregate Domain Analysis (ADA) of these and other interactions were created using SIPMeta[29]. TAD identification, aggregate TAD analysis (ATA), and insulation scores were obtained from Genova[76] at 10 kb resolution.

**ATAC-seq and RNA-seq data**. ATAC-seq and nuclear RNA-seq data on sorted vHIP neurons from dioestrus, proestrus and male groups were previously generated in triplicates[14] and are available from the NCBI Gene Expression Omnibus database under accession number GSE114036.

**Integration of Hi-C, ATAC-seq, ChIP-seq, and RNA-seq data**. ATAC-seq and nuclear RNA-seq previously processed and mapped to mm10[14] was used in conjunction with ngs.plot[77] to create average expression profiles as TSSs. Motifs were obtained by meme-chip[75]. Genome browser tracks display bins per million mapped reads (BPM) normalised signal and were visualised alongside arc-plots of interactions using the WashU Epigenome Browser[78]. Gene expression was evaluated by Stringtie[79] to provide transcripts per million (TPM) normalised values. X escapee genes were identified as those with TPM ≥ 1 in females and ≥1.5-fold difference to males. Gene ontology and pathway analyses were performed by EnrichR[80] and the Ingenuity-Pathway Analysis (IPA) software (https://digitalinsights.qiagen.com/products-overview/discovery-insights-portfolio/analysis-and-visualization/qiagen-ipa). Comparison of expression in other tissues was done using the Human Protein Atlas database[34] and for disease models using the Mouse neurological disorders RNA-seq portal[43]. As a proxy for enhancers, we used ATAC-seq peaks that are located >10 kb from any TSS. We validated that these have marks of active enhancers by plotting the H3K4me1 and H3K27ac ChIP-seq signal derived from sorted (NeuN+) CA1 hippocampal neurons (GSM1939123 & GSM1939156)[32].

**Tissue preservation for histology**. Following whole brain isolation, brains were washed with ice cold 0.1 M PBS and fixed in 4% PFA in 0.1 M PBS at 4 °C for 24 h. After fixation, brains were rinsed in cold 0.1 M PBS and underwent sucrose preservation, which involved placing the brains in solutions containing 15% then 30% sucrose dissolved in 0.1 M PBS at 4 °C for 24 h and 48 h, respectively. Brains were then frozen in dry ice-cooled hexane and stored at −80 °C until sectioning. Cryosectioning was performed by embedding brains in optimal cutting temperature compound (OCT) and cutting serial sections on a rotary cryostat (Leica CM1850, Leica Biosystems GmbH). 5–10 μm coronal sections containing the ventral hippocampus were collected on Super Frost Ultra Plus slides (Fisher Scientific) and were processed for either fluorescence in situ hybridisation (FISH) or immunofluorescence microscopy.

**Fluorescence in situ hybridisation (FISH)**. Cryopreserved brain sections from $n = 3$ animals/group were sent to Albert Einstein College of Medicine Cytogenetics Core Facility for FISH. Briefly, four bacterial artificial chromosome (BAC) clones corresponding to the regions of interest were obtained from the BACPAC Resources Centre (Children's Hospital Oakland Research Institute). DNA was isolated from bacterial clones, labelled with fluorophores, and hybridised to tissue sections, as previously described[81]. Images were acquired using a Zeiss Axiovert 200 inverted microscope (Carl Zeiss MicroImaging, Inc.) using fluorophore-specific filters. The targeted regions were as follows (mm10): Probe 1—chrX: chrX:153798639-153942772 (BAC RP24-305G7, Red label); Probe 2—chrX: 143115199-143348205 (BAC RP24-88L14, Aqua label); Probe 3—chrX: 148937622-149119722 (BAC RP24-295N13, FITC label); Probe 4—chr1: 133514089-133681697 (BAC RP24-287A12, Yellow label). Data analysis of FISH images was performed using ImageJ v1.53 (public domain software from the National Institutes of Health; http://imagej.nih.gov/ij/). For each X chromosome the distance between the centre of the red and blue signals (RB) and the red and green signals (RG) was measured. The difference between these measurements (RB-RG) was calculated for each nucleus in each group and data analysis was performed using one-way ANOVA with Tukey as a post hoc test. The analysis was restricted to X chromosomes positive for all three probe signals ($n = 116–163$/ group for the ventral hippocampus; $n = 190–244$/group for the visual cortex).

**Immunofluorescence microscopy**. Slides with cryopreserved brain sections were rehydrated in PBS and underwent blocking in 0.25% Triton X-100 and 5% BSA in PBS for 1 h at room temperature. Following blocking, the slides were incubated at 4 °C for 24 h with the following primary antibodies: (I) rabbit polyclonal anti-ERα antibody (1:2000; MilliporeSigma, 06-935) (II) mouse monoclonal anti-NeuN conjugated to AlexaFluor-488 (1:500; MilliporeSigma, MAB377X). Slides were subsequently incubated in the dark for 2 h at room temperature with secondary antibody donkey anti-rabbit IgG conjugated to AlexaFluor-594 (1:250; Invitrogen, A-21207). After washing with PBS, the slides were counterstained with DAPI (1:1000) and mounted with Mowiol 4-88 mounting medium (MilliporeSigma). Imaging was performed using a Leica TCS SP8 confocal microscopy system (Leica Microsystems CMS GmbH).

**Elevated Plus Maze**. Oestradiol- or vehicle-injected dioestrus or ovariectomized female mice were subjected to the elevated plus maze test 4 h post-injection. The elevated plus maze apparatus (Stoelting) is a raised (50 cm tall) plus-shaped platform with a set of two open arms (35 × 5 cm) and a set of two closed arms (35 × 5 cm) that are protected by walls (15 cm high). Each mouse began the test by being placed in the centre area that connects the open and closed arms. Animals were allowed to freely explore the maze for 5-min, during which time we recorded time spent in the open arms, time spent in the closed arms, and the total distance travelled. The data were analysed using two tailed $t$ test in R software.

**Reporting summary**. Further information on research design is available in the Nature Research Reporting Summary linked to this article.

## Data availability

The data that support this study are available from the corresponding authors upon reasonable request. Hi-C data generated in this study are available from the NCBI Gene Expression Omnibus (GEO) database under accession number GSE172228.

ATAC-seq and nuclear RNA-seq data on sorted vHIP neurons from dioestrus, proestrus and male groups that we previously published[14] are available from the GEO database under accession number GSE114036.

The H3K4me1 and H3K27ac ChIP-seq data used to validate enhancers is available from the GEO database under accession numbers GSM1939123 and GSM1939156, respectively.

The Hi-C and H3K27me3 ChIP-seq data from ES, NPC, and CN is available from GEO accession GSE96107 and from ENCODE accession ENCSR059MBO. The mm10 genome assembly is available at https://www.ncbi.nlm.nih.gov/assembly/GCF_000001635.20/. Source data are provided with this paper.

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

## Acknowledgements

This work was supported by the National Institutes of Health: the National Institute of Mental Health under Award Number R01MH123523 (to M.K.) and the National Institute of General Medical Sciences under Award Number R00GM127671 (to M.J.R.). S.C.F. holds the Kim B. and Stephen E. Bepler Professorship in Biology. We would like to

thank: Yu Zhang for his assistance with nuclei sorting; Jidong Shan and Cristina Montagna for their assistance with FISH; Samuel Phillips for his assistance with massively parallel sequencing; and Tony Leesnitzer for his assistance with oestradiol quantification.

## Author contributions

M.K. conceived and designed the study; M.J.R. and M.K. directed the project and provided funding; D.R. performed all animal work for the oestrous cycle experiments, immunocytochemistry, tissue preparation for FISH, assisted with nuclei prep and sorting for Hi-C, and performed all behavioural experiments; D.R. and L.O. performed oestrogen injections as well as brain tissue and blood collection for oestrogen replacement experiments; M.K. isolated neuronal nuclei, performed the Hi-C assay, and constructed Hi-C libraries; M.S. and M.J.R. processed the Hi-C data, including mapping, quality filtering, and comparison of replicates, as well as the identification and analysis of compartments, CTCF loops, and E–P interactions; M.S. and M.J.R. performed reprocessing of the ChIP-seq, ATAC-seq, and RNA-seq data and comparison to Hi-C; A.K. performed gene enrichment pathway analyses; D.R. performed the analysis of FISH and behavioural data; S.C.F. performed confocal microscopy; D.R., M.S., S.C.F., M.J.R. and M.K. interpreted the data and constructed the figures; M.K. wrote the article with contributions from all co-authors. D.R., M.S., M.J.R. and M.K. revised the paper.

## Competing interests

The authors declare no competing interests.
