## [Peer Review File · Nature Communications]

REVIEWER COMMENTS

Reviewer #1 (Remarks to the Author):

In this manuscript Rocks et al. provide evidence that 3D chromatin structure in the brain differs between males and females mice and undergoes dynamic remodeling during the female oestrous cycle.

This line of research expands their previous findings (PMID: 31253786) where the same group characterized the effect of the oestrous cycle and sex on neuronal chromatin accessibility and gene expression. The new data shows that chromatin conformation undergoes substantial changes at multiple levels of chromatin organization such as A/B compartments and chromatin loops. This is a novel and original contribution to the chromatin field since sex differences and sex hormone-mediated changes in the higher-order chromatin organization in the brain has been understudied until now. By performing genome-wide HiC and FISH for selected loci in male versus female (both in the proestus and dioestrus stages), and integrating these results with ATAC-seq and RNA-seq data, Rocks et al findings reveal unique 3D genome dynamics in the female brain that have the potential to contribute to both brain and behavioral plasticity and female-specific risks associated with the oestrous cycle such as anxiety and depression.

The manuscript is very well written and organized and I believe this work represents a nice contribution to the field of chromatin conformation.

There are however a few comments that need to be addressed:

1. Throughout the manuscript the authors refer to CTCF loops, that were called using SIP (PMID: 32127418). However, there are a variety of computational methods to look at topologically associated domains (TADs), which represent an important layer of chromatin organization. Rocks et al could use matrices in 20kb resolution to identify TADs (for example with hicFindTADs), and check for changes in TAD boundaries and insulation score (PMID: 29416042) among their different samples and/or at least verify that their overall conclusions still hold true also at the TADs level.

2. Concerning chromosome territories, the authors refer to the limited degree of HiC trans-interactions to support the notion that there are distinct chromosome territories in neurons. Since this finding does not add any novelty to the literature I would not put too much emphasis on this aspect of their research, and maybe remove it from line 77 and Figure 1b, since this data is not relevant for the purpose of their analysis.

3. Figure 1J: How do the authors define enhancer regions? I might have missed it, but otherwise this information should be included in the manuscript. Was ATAC-seq signal +/- 2Kb from TSS considered as enhancer signal? Do the authors have H3K27ac ChIP-seq data available in their cells of interest? In addition to Figure 1J, it would be informative to look also at enhancer connectivity (one enhancer connected to multiple genes) and use RNA-seq data to determine gene co-regulation (and perform gene ontology of genes linked to the highest connected enhancers).

4. In Figure 3 Rocks et al. compare CTCF loops in "die female vs male" and "combined-female vs male". What about "pro female vs male"? If no significant difference is observed please mention it and/or include it in supplemental figures.

5. In relation to Figure 3e, can the authors speculate on the observed differential expression of genes outside of CTCF loop anchors? How can this be explained?

6. Regarding promoter-enhancer loops: it is quite accepted in the field that HiC is not ideal to call prom-enh loops, and especially not at the resolution used by Rocks et al. Have the authors considered performing H3K27ac HiChIP?

7. The authors speculate that the lack of gene expression changes of relevant genes whose connectivity is altered may represent an "epigenetic priming event" and that maybe changes in gene expression would require another stimulus such as stress or other risk factors. This theory is intriguing and it would be very interesting to check if indeed the female brain is "primed" to express those genes under a proper stimulus. Is this kind of experiment feasible for the current manuscript or as a follow up experiment?

Minor comments:

8. Figure legends: instead of stating conclusions, figure legends should just describe the data reported (leaving the interpretation in the main text of the manuscript). Please correct this in Figure legend 1 (h to k) and in the other fig legends where relevant.

9. In Figure 5 (d-i) it is very difficult to differentiate the colors of die and pro loops. Please use a different tonality to highlight the die and pro loop arcs.

Reviewer #2 (Remarks to the Author):

In this manuscript, the authors describes significant multi-level changes in 3D genome dynamics in the female brain across the oestrous cycle. In the previous study done by the same research group, the authors demonstrated sex-specific chromatin dynamics showing that chromatin accessibility varies with the oestrous cycle and sex in the ventral hippocampus. This study is an extension of their previous study, but digging into more molecular detail by mapping 3D genome organization in ventral hippocampal neurons across the oestrous cycle and by sex in mice. They applied and integrated diverse molecular biological tools such as unbiased Hi-C, FISH, ATAC-seq and RNA-seq on the same biological samples. They found estrous cycle-driven dynamism in 3D genome organization, particularly in oestrogen response elements-enriched X chromosome compartments, autosomal CTCF loops, and enhancer-promoter interactions. Interestingly, with high oestrogen levels, the female 3D chromatin becomes similar to the male 3D chromatin. Experimental design and data interpretation are sound and the manuscript is well written, but I have some concerns that need to be improved before publication.

Major points:

1. They used ventral hippocampus for the analysis. I wonder whether all these sexual dimorphic and cycle dependent changes are specific only to vHIP neurons. What about other brain areas such as cortical neurons? Other types of cells or astrocytes? Excitatory neurons vs. inhibitory neurons?

2. To show more specifically estrous cycle-driven dynamism, they have to use ovariectomized females as control, which is missing in the current study.

Minor points

1. The authors discussed the role of CTCF in synaptic plasticity and cognition, but the following references were not referred.

-Kim S et al. J Neurosci (2018) 38:5042; Choi DI et al. Mol Brain (2021) 14:1

2. Typos:

-line 183, Fig 3h-i (Fig 2h-i?)

-line 255, which similar to..

Reviewer #3 (Remarks to the Author):

Summary

Rocks and colleagues investigate the differences in 3D-genome architecture between sex and oestrous-cycle phases (proestrus vs diestrus) using Hi-C profiling and ventral hippocampus samples, as a continuation of a previous publication (Jaric et al., 2019 Nat Commun, PMID: 31253786) in which authors reported significant differences in chromatin organization (i.e., ATAC-seq) and RNA transcription (i.e., RNA-seq) using the same set of samples.

They now provide an extra layer of information (i.e., Hi-C) that is integrated with the previous data (i.e., ATACseq and RNAseq) to investigate sex- and oestrous- chromatin changes at higher structure level: chromosomal compartmentalization, CTCF loops, and enhancer-promoter interactions.

In line with their previous publication, most of differences arise from the male vs female in the diestrus state comparison. However, the magnitude and spreading of these differences seem to decrease with the level of chromatin configuration (i.e., ATAC-seq peaks > DNA loops > chromosomal compartments).

Comments

1-Authors report no significant differences at the level of chromosomal compartmentalization between proestrus and diestrus, and male and diestrus comparisons but for the X chromosome. However, Supplementary Figure 2 also shows a drop in eigen-correlation in chromosome 14, which shares sex-dimorphic features with sexual chromosomes (PMID: 27795737). This point should be followed up and/or better discussed in the manuscript.

2-They also report similar change-directionality between males and proestrus when compared with the diestrus state. These changes seem to be enriched for oestrogen receptor motifs. However, the statistical significance, directionality and magnitude of this enrichment are not provided. Oestrogen is believed to instruct chromatin configuration (PMID: 23684889) and, accordingly, a higher DNA-binding and chromatin-change would be expected for the proestrus state. Authors should better describe this finding and, if it is the case, clarify and/or discuss this apparent contradiction. In addition, an in vitro or in vivo experimental validation of the identified chromatin dynamics/effectors, such as top candidate "oestrogen" changes (e.g., oestrogen treated vs non-treated), would increase the confidence of the results.

3-The manuscript would also benefit of a deeper analysis of chromatin structure that include topological associating domains (TADs), CTCF loop motif-orientation, CTCF-loop co-occurrence with other transcription factors, inter- and intra- CTCF loop interactions, etc.

4-Authors use "fit-Hi-C2 signals with ATAC-seq peaks" as a proxy for enhancers to investigate enhancer-promoter interactions, which is a rather loose criteria. Only a small fraction of ATAC peaks and promoter-interacting regions represent actual enhancers (PMID: 25938943, PMID: 25752748). Authors could take advantage of similar publications in which active enhancers marks are provided, such as acetylation of H3 at lysine 27 (H3K27ac), to refine their analysis (PMID: 31501571, PMID: 32451484). Also, a more thorough

comparison with previous literature, when possible (e.g., male 3D genome architecture with PMID: 25938943, PMID: 25752748,) will be helpful to judge and compare the current data with the previous studies.

5- Authors might also consider to isolate Xa and Xi components using the ATAC-seq data. ATAC-seq peaks are mainly associated with the Xa chromosome, with the exception of chromosome inactivation escapee genes in the Xi chromosome (PMID: 27437574). The analysis of 3D interactions on ATAC-seq positive regions should allow to investigate Xa-specific changes, paying particular attention to escapee genes and ATAC-seq changing regions.

6-The manuscript will benefit of a better justification of the selected examples to illustrate the observed effects (e.g., top ranked genes, etc).

7-For the sake of clarity, the term "significant" should only be used when comparisons are supported by statistical tests and, if so, the statistical test and value should be clearly indicated.

8-For completeness, the APA score of enhancer-enhancer and promoter-promoter interactions should also be provided.

9-Besides the quantification of enhancer-promoter interactions, a further description of the number of genes affected and their chromosomal location will also help to understand the magnitude and extension of the phenomenon.

10-Reference to (Fig. 3h-i), page 10, line 186, do not match.

11-High (i.e., compartments) and middle level (i.e, DNA loops) 3D chromatin changes, besides of lower chromatin changes (i.e., ATAC-seq peaks, Jaric et al., 2019 Nat Commun, PMID: 31253786), do not seem to be aligned. This apparent lack of coherence, and its functional consequences, should be better discussed in the manuscript.

Response to Reviewer's Comments

We would like to thank the reviewers for their time and effort as well as for thoughtful comments and suggestions which, we believe, resulted in much improved manuscript. As per editor's and reviewers' requests, we have performed a comprehensive manuscript revision which include additional experiments (Hi-C, FISH, hormone quantification, behavioral analyses) and multiple additional data analyses. To address the brain region specificity of the estrous cycle- and sex-driven 3D genome changes, we performed the *FISH* assay in the visual cortex. To provide a mechanistic insight, more specifically to address the role of estrogen in the estrous cycle-driven 3D chromatin changes, we performed Hi-C assay on sorted vHIP neurons isolated from estrogen- and vehicle-treated ovariectomized (OVX) female mice. We ran two additional animal cohorts for behavioral analyses – to ensure that the estrogen dose regimen induces physiological changes in behavior and to test the OVX animals' responsiveness to estrogen. We also addressed all bioinformatics inquires including: the definition of enhancers and enhancer-promoter interactions, the inquiry related to TAD and hierarchical model of 3D genome organization, and sexual dimorphism in chromosome 14, among others.

Our revised manuscript includes: two new figures in the main text (*Figures 6 and 7*); two full supplementary figures (*Supplementary Figure 3 and 9*); multiple additions to the existing supplementary figures (*Suppl. Figure 1c-f, Suppl. Figure 2e-f; Suppl. Figure 4e; Suppl. Figure 5a-c, e-g, j-l*); two supplementary tables (*Supplementary Tables 4-5*); and about 3,400 words of the added text (marked by blue font color). Below please find a point-by-point response to all reviewers' comments. The text was also revised accordingly.

Reviewer #1

- We would like to thank the reviewer for the careful analysis as well as for thoughtful and kind comments and suggestions which have significantly improved our manuscript. Please find below responses to all your comments.

In this manuscript Rocks et al. provide evidence that 3D chromatin structure in the brain differs between males and females mice and undergoes dynamic remodeling during the female oestrous cycle. This line of research expands their previous findings (PMID: 31253786) where the same group characterized the effect of the oestrous cycle and sex on neuronal chromatin accessibility and gene expression. The new data shows that chromatin conformation undergoes substantial changes at multiple levels of chromatin organization such as A/B compartments and chromatin loops. This is a novel and original contribution to the chromatin field since sex differences and sex hormone-mediated changes in the higher-order chromatin organization in the brain has been understudied until now. By performing genome-wide HiC and FISH for selected loci in male versus female (both in the proestus and dioestrus stages), and integrating these results with ATAC-seq and RNA-seq data, Rocks et al findings reveal unique 3D genome dynamics in the female brain that have the potential to contribute to both brain and behavioral plasticity and female-specific risks associated with the oestrous cycle such as anxiety and depression.

The manuscript is very well written and organized and I believe this work represents a nice contribution to the field of chromatin conformation.

There are however a few comments that need to be addressed:

1. Throughout the manuscript the authors refer to CTCF loops, that were called using SIP

(PMID: 32127418). However, there are a variety of computational methods to look at topologically associated domains (TADs), which represent an important layer of chromatin organization. Rocks et al could use matrices in 20kb resolution to identify TADs (for example with hicFindTADs), and check for changes in TAD boundaries and insulation score (PMID: 29416042) among their different samples and/or at least verify that their overall conclusions still hold true also at the TADs level.

- We added the requested analysis of TADs including Aggregate TAD Analysis (ATA) and tornado plots of insulation scores at 10 kb resolution using Genova. Interestingly, we see no obvious differences in these features (Supplementary Figure 5e). We then more specifically examined insulation and domain formation at differential loops, but again saw no differences in domain or insulation scores (Supplementary Figure 5f,g,j,k). Investigating further, we found overrepresentation of CTCF sites in tandem orientation at differential loop anchors, despite unchanged loops being overwhelmingly in convergent orientation (Supplementary Figure 5c). This is in line with tandem loops not being as important for domain formation (Tang et al. Cell 2016; Rowley et al., Genome Research 2020). We also found that many differential loops share anchors with unchanged loops (i.e. are “nested” loops), which is likely why insulation is preserved at these sites (Supplementary Figure 5l). Importantly, the lack of changes to domains and insulation indicates greater specificity to these loops, explaining the lack of expression changes for genes in the loop interior. Rather, we posit that these differential loops may prime nearby genes for responsiveness, similar to enhancer-promoter interactions. We thank the reviewer for turning our focus in this direction as it provided valuable insights into the relationship between these differential loops and gene expression. We’ve added the following text to the manuscript:

(Lines 252-265): *“To understand this further, we first performed genome-wide Aggregate TAD Analysis (ATA) as well as plotted insulation scores at their borders, but found no evident differences in these domain signals or insulation (**Supplementary Fig. 5e**). To more specifically investigate how these differential CTCF loops may impact insulated loop domains, we examined insulation scores and performed aggregate domain analysis (ADA) at differential loops. Curiously, changes in the loops did not result in obvious differences in insulation scores or loop domains (**Supplementary Fig. 5f-g**). This is consistent with these differential loops representing non-convergent loops which are less important for interaction domains^{29,37}, but acting more precisely similar to enhancer-promoter interactions. This finding is also consistent with recent evidence suggesting that CTCF loops are likely only a small component of interaction domains and gene expression control^{30,38}. This may be the reason why we see no genome-wide changes to the expression of genes that are interior to differential loops (**Supplementary Fig. 5d**) but do detect changes in expression of *Errb4* and *Mreg*, which are next to differential loop anchors but are located outside the loop (**Fig. 3e**).”*

(Lines 279-285): *“Similar to what we found for sex-specific loops, there was, in general, little correlation between differential Pro-Die loops, insulation scores, loop domains, and gene expression (**Supplementary Fig. 5i-k**). The lack of change to insulation or domain structure is likely due to the same reasons we saw lack of changes in these features in the Die-Male comparison as mentioned above. We also note that many of these loops are “nested” in that they share loop anchors with other loops that are unchanged between samples which likely stabilizes insulation at these anchors (**Fig. 4f-g, Supplementary Fig. 5l-m**).”*

(Lines 469-484) *“Intriguingly, the oestrus cycle- and sex-driven compartment changes on the X chromosome are not composed of altered CTCF loops, and therefore do not support the hierarchical model of chromosome organization. Instead, our results are more consistent with a model where compartments and CTCF loops are independent layers of genome organization¹⁶.*

This model has been supported by studies where degradation of looping factors results in little to no change in compartments^{30,54}. These earlier studies also showed that de novo identification of chromatin contact domains by Hi-C, sometimes referred to as topologically associating domains (TADs), often represents a mixture of CTCF loop domains and compartment domains^{16,30,55,56}. The definition of “TADs” is of interest, but is discussed elsewhere and is not the purpose of this study⁵⁶. However, we acknowledge that, in recent years, the term “TAD” is becoming more synonymous with CTCF loop domains. Curiously, despite differences in CTCF loops, we do not detect changes in contact domains, i.e. Genova-identified “TADs” nor in CTCF loop domains. This is likely because cycle-specific loops tend to share one or more anchors with unchanging loops. Therefore, the oestrous cycle-driven changes to CTCF loops are likely important for priming the regulation of specific genes nearby, rather than the larger numbers of genes found inside loop domains.”

2. Concerning chromosome territories, the authors refer to the limited degree of HiC trans-interactions to support the notion that there are distinct chromosome territories in neurons. Since this finding does not add any novelty to the literature I would not put too much emphasis on this aspect of their research, and maybe remove it from line 77 and Figure 1b, since this data is not relevant for the purpose of their analysis.

- We agree with the reviewer that we did not specifically analyze chromosome territories in our study. In response to this comment, we removed “chromosome territories” from line 77 and Figure 1b, as requested. We kept the scheme of nucleus/chromosomes just to orient the reader that the three levels of organization are within each chromosome. This sentence now reads as follows:

(Lines 77-79): “We performed bioinformatics analysis on Hi-C libraries focusing on three different levels of 3D chromatin organization within each chromosome: chromosome compartments, CTCF loops and loop domains, and enhancer-promoter (E-P) interactions (Fig. 1b).”

3. Figure 1J: How do the authors define enhancer regions? I might have missed it, but otherwise this information should be included in the manuscript. Was ATAC-seq signal +/-2Kb from TSS considered as enhancer signal? Do the authors have H3K27ac ChIP-seq data available in their cells of interest? In addition to Figure 1J, it would be informative to look also at enhancer connectivity (one enhancer connected to multiple genes) and use RNA-seq data to determine gene co-regulation (and perform gene ontology of genes linked to the highest connected enhancers).

Thank you for raising this important point; this, indeed, was not clearly stated in the manuscript and we now included this information. We define enhancers as sites with ATAC-seq peaks more than one 10 kb bin from a TSS. We found H3K27ac and H3K4me1 ChIP-seq data from sorted hippocampal neurons and examined the signal at our putative enhancers. We have added a heatmap of this signal and average profiles, demonstrating that our “putative enhancers” have H3K4me1 and H3K27ac as well (**Supplementary Fig. 1e**). This is also added to the main text:

*(Lines 115-119): “For this purpose, we used non-promoter ATAC-seq peaks, ≥ 10 kb from transcription start sites (TSSs), as a proxy for enhancers. To justify our choice, we show that published H3K4me1 and H3K27ac data derived from sorted hippocampal neurons³² identify these peaks as loci with the marks of active enhancers (**Supplementary Fig. 1e**), consistent with our previous findings in cortical neurons³³.”*

(Lines 724-727): “As a proxy for enhancers, we used ATAC-seq peaks that are located > 10 kb from any TSS. We validated that these have marks of active enhancers by plotting the H3K4me1 and H3K27ac ChIP-seq signal derived from sorted (NeuN+) CA1 hippocampal neurons (GSM1939123 & GSM1939156)³².”

4. In Figure 3 Rocks et al. compare CTCF loops in “die female vs male” and “combined-female vs male”. What about “pro female vs male”? If no significant difference is observed please mention it and/or include it in supplemental figures.

Upon performing the requested analysis, we found evidence for approximately 280 loops that are higher in Proestrus than Males, which includes the 100 differential loops common to females (Fig. 3a), and an additional 180 loops that are Proestrus only (Pro high, while Male & Die low), (Suppl. Figure 5a). We added these figure panels and now discuss these results in the main text.

However, we wish to note that loop calling in Hi-C is naturally prone to a degree of statistical error (see Rowley et al., Genome Research 2020; Also see our recent preprint Gu et al., bioRxiv 2021). Performing multiple comparison analysis (i.e. 3-way) will likely expand these errors. To combat this, we primarily use statistical thresholds for identification of loops in each sample (SIP), and then, for differential analysis, rely on a pairwise comparison of the actual loop signal, taking a stringent fold-change threshold. Therefore, there are likely many loops with smaller changes that do not meet our threshold. Indeed, our analysis indicates that Proestrus loop signal is often in between that of Dioestrus and Males (Suppl. Figure 5b, Fig. 4C). We added the following to the manuscript:

(Lines 231-233): “We found an increased ability (1.65 times) to call sex-specific loops when comparing dioestrus to males, as opposed to comparing mixed females to males (**Fig. 3a, Supplementary Fig. 5a**)”

(Lines 237-239): “Loop signal in proestrus was between that of dioestrus and males for both categories of loops, but also with many loops specific to proestrus (**Supplementary Fig. 5b**) which we examine later (**Fig. 4**).”

(Lines 270-274) “Of the Pro-specific loops, 84 were called differential in the male to die comparison (male > die), leaving 180 loops that were proestrus only (**Supplementary Fig. 5a**). However, when we compared their signal intensity to dioestrus, the loop signal changes in proestrus were largely in the same direction as the loop signal changes in males, simply to a different degree (**Fig. 4c**).”

5. In relation to Figure 3e, can the authors speculate on the observed differential expression of genes outside of CTCF loop anchors? How can this be explained?

To address this question, we performed a set of new analyses showing that differential CTCF loops occur at non-convergently oriented motifs (**Supplementary Fig. 5c**), with little effect on domain or insulation scores (**Suppl. Fig. 5e-g**). Based on the new results, we now posit that these loops are acting more analogously to enhancer-promoter interactions as others have suggested for tandem CTCF loops (Tang et al., Cell 2016). We have added this explanation to the main text and suggested an explanation for the observed differential expression of genes outside of CTCF loop anchors (concerning Figure 3e), as the reviewer suggested.

(Lines 240-241): “Intriguingly, CTCF motifs at differential loops were frequently in non-convergent orientations (**Supplementary Fig. 5c**).”

(Lines 252-260): “To understand this further, we first performed genome-wide Aggregate TAD Analysis (ATA) as well as plotted insulation scores at their borders, but found no evident differences in these domain signals or insulation (**Supplementary Fig. 5e**). To more specifically investigate how these differential CTCF loops may impact insulated loop domains, we examined insulation scores and performed aggregate domain analysis (ADA) at differential loops. Curiously, changes in the loops did not result in obvious differences in insulation scores or loop domains (**Supplementary Fig. 5f-g**). This is consistent with these differential loops representing non-convergent loops which are less important for interaction domains^{29,37}, but acting more precisely similar to enhancer-promoter interactions.”

(Lines 262-265): “This may be the reason why we see no genome-wide changes to the expression of genes that are interior to differential loops (**Supplementary Fig. 5d**) but do detect changes in expression of *Erbb4* and *Mreg*, which are next to differential loop anchors but are located outside the loop (**Fig. 3e**).”

6. Regarding promoter-enhancer loops: it is quite accepted in the field that HiC is not ideal to call prom-enh loops, and especially not at the resolution used by Rocks et al. Have the authors considered performing H3K27ac HiChIP?

We agree that H3K27ac HiChIP would be an excellent approach to better identify and understand enhancer-promoter interactions. Unfortunately, the HiChIP protocol requires millions of cells to generate high quality data, a number we cannot obtain for this cell type. To remind the reviewer, we work with sorted neuronal nuclei from a specific brain subregion, and we get 200,000 nuclei when we combine material from two animals.

Therefore, while HiChIP would provide valuable extra information, it is not currently feasible for us with this cell type. Regarding the utility of Hi-C for this purpose, many studies have used Hi-C at 10 kb resolution (or even coarser) to analyze enhancer-promoter interactions (Hu et al., Nat Comm 2021, PMC8233376; Chen et al., Nat Comm 2020 PMC7156422; Lu et al., Mol Cell 2020, PMID:32592681). Indeed, FitHiC2 (which we use) was created to identify these types of interactions from Hi-C data (Kaul et al., Nat. Prot. 2020). We considered changing the name of these interactions from E-P, but think that it still best describes interactions which are identified as significant by FitHiC2, and connect an enhancer and promoter on either side. Therefore, we discuss them as enhancer-promoter interactions. We hope the reviewer understands our efforts to be clear on this issue for the reader. As such, we added the following statement to the Discussion section:

(Lines 499-502): “While our E-P results were obtained by examining significant interactions in Hi-C data, future work using methods that are specific for E-P interactions, such as HiChIP⁶² for H3K4me1 or H3K27ac, may identify additional sex specific- or sex hormone-driven E-P interactions critical for gene regulation and neuronal function.”

7. The authors speculate that the lack of gene expression changes of relevant genes whose connectivity is altered may represent an “epigenetic priming event” and that maybe changes in gene expression would require another stimulus such as stress or other risk factors. This theory is intriguing and it would be very interesting to check if indeed the female brain is “primed” to express those genes under a proper stimulus. Is this kind of experiment feasible for the current manuscript or as a follow up experiment?

- Thank you for this kind comment. We are also very intrigued by the possibility of hormonal epigenetic priming. While these experiments are out of scope of the current manuscript, this is

going to be one of the most important directions for our follow-up studies.

Minor comments:

8. Figure legends: instead of stating conclusions, figure legends should just describe the data reported (leaving the interpretation in the main text of the manuscript). Please correct this in Figure legend 1 (h to k) and in the other fig legends where relevant.

We corrected the legend as requested. It now reads:

(Lines 1016-1018): *“As a third component of chromatin organization, enhancer-promoter (E-P) interactions are shown (h) with their average distance normalized Hi-C signal (i), the number of interactions across genes, (j) and their relationship with gene expression (k).”*

9. In Figure 5 (d-i) it is very difficult to differentiate the colors of die and pro loops. Please use a different tonality to highlight the die and pro loop arcs.

- To allow for easy data interpretation, we use the same color codes for diestrus and proestrus throughout the manuscript, so we are not able to change the tonality for this case only. However, we increased the resolution of our figure (and are able to provide even a higher resolution images if needed) so we believe that this now allows for easier color differentiation.

Reviewer #2

- We would like to thank the reviewer for a positive review and great suggestions which improved our study greatly. Please find below a response to your comments.

In this manuscript, the authors describes significant multi-level changes in 3D genome dynamics in the female brain across the oestrous cycle. In the previous study done by the same research group, the authors demonstrated sex-specific chromatin dynamics showing that chromatin accessibility varies with the oestrous cycle and sex in the ventral hippocampus. This study is an extension of their previous study, but digging into more molecular detail by mapping 3D genome organization in ventral hippocampal neurons across the oestrous cycle and by sex in mice. They applied and integrated diverse molecular biological tools such as unbiased Hi-C, FISH, ATAC-seq and RNA-seq on the same biological samples. They found estrous cycle-driven dynamism in 3D genome organization, particularly in oestrogen response elements-enriched X chromosome compartments, autosomal CTCF loops, and enhancer-promoter interactions. Interestingly, with high oestrogen levels, the female 3D chromatin becomes similar to the male 3D chromatin. Experimental design and data interpretation are sound and the manuscript is well written, but I have some concerns that need to be improved before publication.

Major points:

1. They used ventral hippocampus for the analysis. I wonder whether all these sexual dimorphic and cycle dependent changes are specific only to vHIP neurons. What about other brain areas such as cortical neurons? Other types of cells or astrocytes? Excitatory neurons vs. inhibitory neurons?

- Thank you for this important comment. We plan to do a comprehensive analysis of other brain regions and cell types but this would need to be a separate manuscript. However, to directly address your question, we performed the FISH assay in the visual cortex, with the same probes that we used for the vHIP. Interestingly, these data indicate that the estrous cycle-dependent changes are, indeed, brain region specific as we did not detect a significant difference between proestrus and diestrus in this brain region. We would like to add that this is not surprising to us, as the visual cortex has a lower density of estrogen and progesterone receptors compared to the hippocampus, so the extent of chromatin changes with the cycle is likely to be smaller. Importantly, we did confirm sex difference (basically this is the Male-Diestrus difference in the vHIP), which further adds confidence to this data. The results are presented in the newly added **Supplementary Figure 3**. We have also added the following text in the main manuscript:

(Lines 188-191): *“Interestingly, these estrous cycle-driven changes may be brain region-specific, as the same DNA-FISH assay in the visual cortex did not show a significant difference between proestrus and dioestrus, despite confirming the sex difference in the physical distance among the tested probes (Suppl. Fig. 3).”*

2. To show more specifically estrous cycle-driven dynamism, they have to use ovariectomized females as control, which is missing in the current study.

- We would like to thank the reviewer for this comment. We added the experiments of ovariectomized (OVX) animals with either short-term estradiol or vehicle treatment. The datasets that we added are comprehensive and include not only Hi-C analysis on the two animal groups but complementary behavioral analyses as well. It was important that we were able to re-create, in part, proestrus-specific 3D genome changes with estrogen treatment in OVX animals, which we explain in more detail in our Response to Reviewer #3 (Question #2). However, Reviewer 2's comment was on using OVX females as control for our study of the estrous cycle and we would like to provide a comprehensive response to this specific point.

- We acknowledge that OVX animals are used frequently in neuroendocrinology as they allow for hormone replacement experiments where the role of specific hormones within the complex hormonal milieu of the ovarian cycle can be dissected. However, the removal of ovaries leads to many brain adaptations, on pretty much every level, from sex hormone receptor expression to changes in behavior and response to sex hormones. One of the best examples of this is the so-called “window of opportunity” for hormone replacement therapy (HRT) in postmenopausal women; it is known that if the HRT is not initiated in a timely fashion, the female brain loses its responsiveness to estrogen. While we use young adult animals, after ovariectomy, these animals also undergo big changes within the brain in response to hormonal withdrawal and they are different from cycling animals. We were able to show that ovariectomy leads to both changes in 3D genome and in behavioral responsiveness to estrogen. By itself, we believe this is a great contribution to the fields of neuroendocrinology and chromatin regulation, while at the same time these data highlight the critical importance of using naturally-cycling animals to understand the 3D chromatin dynamics in the female brain in response to fluctuating sex hormone levels. We added **Supplementary Figure 9** and text in both results and in discussion to elaborate on these data.

(Lines 404-426): *“Finally, we note that the overlap between changes in 3D genome organization across the cycle (Fig. 2, 4-5) and those induced by EB treatment in OVX animals (Figs. 6-7), while obvious, it is far from complete and includes 15-25% of overlap across the three levels of organization (Fig. 6d, 7c, 7e). To explore a possible reason for the limited data overlap, we*

down-sampled OVX Hi-C data (**Suppl. Table 4**) to ensure equal number of sequencing reads in all samples and then compared the overall Hi-C signal in cycling animals with that in OVX animals (**Suppl. Fig. 9a**). Importantly, the PCA analysis showed that the overall Hi-C signal is much more similar between dioestrus and proestrus groups compared to that of the OVX samples (both vehicle and EB treated, **Suppl. Fig. 9a**), indicating that neuronal 3D genome organization undergoes large changes following ovariectomy-induced depletion of endogenous sex hormones. Interestingly, the changed responsiveness of 3D genome to oestrogen following ovariectomy is also mirrored in the changed behavioral response. Indeed, while our 4-hour EB treatment increased the time spent in the open arms of the elevated plus maze (a reduced anxiety index) in low-estrogenic cycling female mice (**Fig. 6a**), this EB regimen was not sufficient to change anxiety-like behavior in OVX mice (**Suppl. Fig. 9b**). The OVX mice did have a behavioral response to oestrogen, which included overall lower activity levels following EB treatment (**Suppl. Fig. 9b**), which was not seen in the cycling animals (**Suppl. Fig. 9c**). This data further indicated that OVX mice undergo brain adaptations that change their response to oestrogen compared to the animals undergoing the regular reproductive cycles. [...] Our findings also highlight the importance of studying the physiological oestrous cycle in order to understand the dynamics of the 3D genome in the female brain.”

(Lines 523-528): “Unsurprisingly, consistent with other ovariectomy-induced brain adaptations⁶⁷⁻⁶⁹, the 3D genome in OVX animals shows an altered state and differential responsiveness to estrogen compared to the 3D genome of naturally-cycling females. Actually, these data further confirm that the higher-order chromatin organization in neurons is highly responsive to sex hormone changes in terms of both increased hormonal levels as well as hormonal withdrawal.”

Minor points

1. The authors discussed the role of CTCF in synaptic plasticity and cognition, but the following references were not referred.

-Kim S et al. J Neurosci (2018) 38:5042; Choi DI et al. Mol Brain (2021) 14:1

- We apologize for this omission. We added a new sentence that clearly states the role of CTCF in synaptic plasticity and memory formation including the references suggested by the reviewer. The text now reads as follows:

(Lines 536-539): “For instance, CTCF has been shown to be critical for synaptic plasticity and memory formation^{70,71}. In studies of mice with disruptions in the CTCF loop organizer, it was shown that memory-relevant genes do not exhibit hippocampal gene expression changes under basal conditions but are affected in activity-dependent way²⁴.”

2. Typos:

-line 183, Fig 3h-i (Fig 2h-i?)

Thank you for noticing this typo. We have now corrected this error.

-line 255, which similar to..

We clarified this sentence and this section now reads as follows:

(Lines 297-299): “Another example includes a loop surrounding the *Thbs2* gene (**Fig. 4g**). Similar to *Adcyap1*, the *Thbs2* gene is also an anxiety-related gene⁴¹ that has an oestrogen response element in the promoter and can be regulated by ER α ⁴².”

Reviewer #3

We would like to thank the reviewer for this thoughtful and very helpful review. We addressed all your comments as explained in detail below.

Summary

Rocks and colleagues investigate the differences in 3D-genome architecture between sex and oestrous-cycle phases (proestrus vs diestrus) using Hi-C profiling and ventral hippocampus samples, as a continuation of a previous publication (Jaric et al., 2019 Nat Commun, PMID: 31253786) in which authors reported significant differences in chromatin organization (i.e., ATAC-seq) and RNA transcription (i.e., RNA-seq) using the same set of samples.

They now provide an extra layer of information (i.e., Hi-C) that is integrated with the previous data (i.e., ATACseq and RNAseq) to investigate sex- and oestrous- chromatin changes at higher structure level: chromosomal compartmentalization, CTCF loops, and enhancer-promoter interactions.

In line with their previous publication, most of differences arise from the male vs female in the diestrus state comparison. However, the magnitude and spreading of these differences seem to decrease with the level of chromatin configuration (i.e., ATAC-seq peaks > DNA loops > chromosomal compartments).

Comments

1-Authors report no significant differences at the level of chromosomal compartmentalization between proestrus and diestrus, and male and diestrus comparisons but for the X chromosome. However, Supplementary Figure 2 also shows a drop in eigen-correlation in chromosome 14, which shares sex-dimorphic features with sexual chromosomes (PMID: 27795737). This point should be followed up and/or better discussed in the manuscript.

We thank the reviewer for this insight. We further investigated the differences on chromosome 14 and found that the published sex chromosome-related genes on chr14 (from PMID: 27795737) indeed have larger differences in compartment structure (Added to Supplementary Fig. 2 – now **Suppl. Figures 2e-f**). We have added these results to the main text and a discussion of these points. Intriguingly, estrogen is also able to change compartmental structure of chromosome 14 in OVX animals. So, we also discuss these new data in light of our previous data, as stated below:

(Lines 151-156): *“Chromosome 14 showed some small differences in compartment signal (**Supplementary Figs. 2a, 2e**) which is consistent with reports that chromosome 14 shares epigenetic features with sex chromosomes³⁵. We found that the sex chromosome-related genes previously found on chromosome 14³⁵ have larger differences in compartment structure than other genes on this chromosome (**Supplementary Fig. 2f**), suggesting that sex-specific autosomal genes may have nuanced sex-specific compartmental structures.”*

(Lines 373-377): *“Oestradiol induced fairly large changes in compartments along the X chromosome (**Fig. 6c**) and no changes in autosomes with an exception of chromosome 14 (**Fig. 6c**), which was previously shown to have sexually dimorphic compartment organization (**Suppl. Fig. 2a**), likely through features shared with the X chromosome³⁵.”*

(Lines 440-443): “The largest autosomal compartment differences were on chromosome 14 which has genes that share regulatory features with the sex chromosomes³⁵, indicating that 3D chromosomal organization reflects sexually dimorphic features.”

2-They also report similar change-directionality between males and proestrous when compared with the diestrous state. These changes seem to be enriched for oestrogen receptor motifs. However, the statistical significance, directionality and magnitude of this enrichment are not provided. Oestrogen is believed to instruct chromatin configuration (PMID: 23684889) and, accordingly, a higher DNA-binding and chromatin-change would be expected for the proestrous state. Authors should better describe this finding and, if it is the case, clarify and/or discuss this apparent contradiction. In addition, an in vitro or in vivo experimental validation of the identified chromatin dynamics/effectors, such as top candidate “oestrogen” changes (e.g., oestrogen treated vs non-treated), would increase the confidence of the results.

- We would like to thank the reviewer for this important comment and experimental suggestion. We would like to note that, while estrogen has been believed to instruct chromatin configuration for quite a long time, primarily based on cell culture experiments, no one has ever performed the study either within the context of the estrous cycle or within the brain. No one has ever compared males to females either. What we show is that the magnitude of the effect is pretty much similar in all these comparisons, whether we talk about male-specific or diestrus-specific loops in the Male-Die comparison, or we talk about proestrus- and diestrus-specific loops in the Pro-Die comparison. The fact that we found the signal in females to become more similar to males once estrogen levels rise in proestrus is intriguing, but it is not contradictory. Males and females are hormonally different so high estrogen does not have to make females more different from males; this all depends on the phenotype in question. In fact, in our previous study (Jaric et al, 2019), we found proestrus females to be more similar to males than to diestrus females with regard to anxiety-related behavior, which is of specific interest to us, and therefore the 3D genome data are very aligned with this behavioral phenotype. On the other hand, the estrous cycle has a very complex hormonal profile beyond changes in estrogen levels, so it is of no surprise that diestrus (with its high progesterone) is associated with as many diestrus-specific loops as we find proestrus-specific loops in proestrus.

- However, we are aware that the studies of the estrous cycle are correlational and that hormone replacement experiments will help confirm our results. For this reason, and as per the reviewer’s suggestion, we performed a comprehensive Hi-C study on vHIP neurons isolated from OVX mice that were treated short-term (4 hours) with either estrogen or vehicle. Here we provide evidence that estrogen can change 3D genome in postmitotic neurons in vivo, at all levels of organization, and that in part these changes overlap with changes that we see in high-estrogenic proestrus phase. But as the reviewer will see, we also show that many changes do not overlap between the natural cycle and estrogen-treated OVX animals, for two main reasons: 1) because proestrus is not only characterized by high-estrogen but has a more complex hormonal profile; 2) because the ovariectomy induces adaptive changes in 3D genome. Importantly, we show the changed responsiveness to estrogen at both 3D genome and behavioral level.

Therefore, thanks to the reviewer’s excellent suggestion, our study now has an additional layer where we confirm the role of estrogen level changes in 3D genome re-organization while at the same time highlighting the importance of studying the 3D genome dynamics in the female brain in its naturalistic context.

Please note we now include the MEME significance scores (E-values) for the motifs in the relevant figure legends, and we added two new main Figures 6 and 7; one Supplementary Figure 9; two new Supplementary Tables (4-5) and more than 2,000 words of text to address this question.

3-The manuscript would also benefit of a deeper analysis of chromatin structure that include topological associating domains (TADs), CTCF loop motif-orientation, CTCF-loop co-occurrence with other transcription factors, inter- and intra- CTCF loop interactions, etc.

We thank the reviewer for these suggestions. We have added analysis of TADs, insulation scores, Aggregate Domain Analysis, and CTCF loop motif-orientation. Interestingly, while the majority of our CTCF loop calls are in convergent orientation, the sex-specific CTCF loops are enriched for tandem orientations. It was previously reported that tandem CTCF loops act more similar to enhancer-promoter interactions, and interestingly we see no obvious defect in insulation scores, TAD signal, or loop domains. We have added these results and a discussion of these points as noted below.

(Lines 240-241): *“Intriguingly, CTCF motifs at differential loops were frequently in non-convergent orientations (**Supplementary Fig. 5c**).”*

(Lines 252-260): *“To understand this further, we first performed genome-wide Aggregate TAD Analysis (ATA) as well as plotted insulation scores at their borders, but found no evident differences in these domain signals or insulation (**Supplementary Fig. 5e**). To more specifically investigate how these differential CTCF loops may impact insulated loop domains, we examined insulation scores and performed aggregate domain analysis (ADA) at differential loops. Curiously, changes in the loops did not result in obvious differences in insulation scores or loop domains (**Supplementary Fig. 5f-g**). This is consistent with these differential loops representing non-convergent loops which are less important for interaction domains^{29,37}, but acting more precisely similar to enhancer-promoter interactions.”*

(Lines 262-265): *“This may be the reason why we see no genome-wide changes to the expression of genes that are interior to differential loops (**Supplementary Fig. 5d**) but do detect changes in expression of *Erbb4* and *Mreg*, which are next to differential loop anchors but are located outside the loop (**Fig. 3e**).”*

4-Authors use “fit-Hi-C2 signals with ATAC-seq peaks” as a proxy for enhancers to investigate enhancer-promoter interactions, which is a rather loose criteria. Only a small fraction of ATAC peaks and promoter-interacting regions represent actual enhancers (PMID: 25938943, PMID: 25752748). Authors could take advantage of similar publications in which active enhancers marks are provided, such as acetylation of H3 at lysine 27 (H3K27ac), to refine their analysis (PMID: 31501571, PMID: 32451484). Also, a more thorough comparison with previous literature, when possible (e.g., male 3D genome architecture with PMID: 25938943, PMID: 25752748,) will be helpful to judge and compare the current data with the previous studies.

This was an excellent suggestion. As our rationale, we used ATAC-seq peaks as a mark of putative enhancers, because accessibility has been one of the primary marks used by ENCODE to annotate enhancers: <https://www.encodeproject.org/data/annotations/>). We also previously published a methods paper showing that our ATAC-seq data in sorted cortical neurons identify functional regulatory elements both promoters and enhancers (Rocks et al, 2021). However, to more specifically respond to the reviewer’s comments, we found H3K27ac and H3K4me1 ChIP-seq data from sorted hippocampal neurons and examined the signal at our putative enhancers.

We have added a heatmap of this signal and average profiles, demonstrating that our “putative enhancers” have H3K4me1 and H3K27ac as well (**Supplementary Fig. 1e**). This is also added to the main text:

(Lines 115-119): “For this purpose, we used non-promoter ATAC-seq peaks, ≥ 10 kb from transcription start sites (TSSs), as a proxy for enhancers. To justify our choice, we show that published H3K4me1 and H3K27ac data derived from sorted hippocampal neurons³² identify these peaks as loci with the marks of active enhancers (**Supplementary Fig. 1e**), consistent with our previous findings in cortical neurons³³”.

(Lines 724-727): “As a proxy for enhancers, we used ATAC-seq peaks that are located > 10 kb from any TSS. We validated that these have marks of active enhancers by plotting the H3K4me1 and H3K27ac ChIP-seq signal derived from sorted (NeuN+) CA1 hippocampal neurons (GSM1939123 & GSM1939156)³².”

We think this greatly supported our claims and thank the reviewer for this suggestion.

5- Authors might also consider to isolate Xa and Xi components using the ATAC-seq data. ATAC-seq peaks are mainly associated with the Xa chromosome, with the exception of chromosome inactivation escapee genes in the Xi chromosome (PMID: 27437574). The analysis of 3D interactions on ATAC-seq positive regions should allow to investigate Xa-specific changes, paying particular attention to escapee genes and ATAC-seq changing regions.

- As suggested, we changed the analysis to separately examine loci on the X chromosome with ATAC-seq peaks. Indeed, these loci had larger changes in compartments, especially at differential ATAC-seq peaks. However, differential ATAC-seq signal cannot fully explain compartment changes, as altered compartments at escapee genes do not have differential ATAC-seq peaks. We now discuss these points in the main text.

(Lines 205-208): “Curiously, we did not find differential ATAC-seq peaks at the promoters of escapee genes, suggesting that differences in compartmental interactions, rather than accessibility, better explains the escape of the genes from dosage compensation (**Supplementary Fig. 4e**)”

6-The manuscript will benefit of a better justification of the selected examples to illustrate the observed effects (e.g., top ranked genes, etc).

- We selected the examples based on the following criteria: the magnitude of effect/change (top ranked genes), the location (for the proximity to the FISH probe) the importance of a gene for our phenotypes of interest (anxiety/depression, estrogen targets), and associated gene expression changes.

We discuss this in the manuscript as shown for the following examples:

(Lines 196-199): “Taking **the top 5 escapee genes** from the Die-Male comparison (**Fig. 2h, Supplementary Fig. 4a**), we indeed found that these genes are always located in the A compartment in dioestrus (**Supplementary Fig. 4b**) and show a higher eigenvector A compartment signal compared to expression matched non-escapee genes (**Fig. 2i, Supplementary Fig. 4c-d**).”

(Lines 205-208): “We also checked the *Htr2c* gene which is **located nearby the FISH probes** that we designed (**Fig. 2e**) and is another gene that is differentially expressed in the Die-Male but not in the Pro-Male comparison¹⁴ (**Supplementary Fig. 4e**).”

(Lines 286-288): “An interesting example is a sex-specific and oestrous cycle-dependent loop involving *Adcyap1*, an important **stress- and oestrogen-sensitive gene implicated in anxiety**^{39,40} (Fig. 4f).”

(Lines 320-323): “For instance, the *Pou3f2* gene, an important **psychiatric risk-related gene encoding a brain-specific transcription factor**, showed clear oestrous cycle-dependent (Fig. 5d) and sex-specific (Supplementary Fig. 6d) E-P interaction profiles which were associated with differential gene expression.”

7-For the sake of clarity, the term “significant” should only be used when comparisons are supported by statistical tests and, if so, the statistical test and value should be clearly indicated.

- We agree and have altered the text accordingly.

8-For completeness, the APA score of enhancer-enhancer and promoter-promoter interactions should also be provided.

- We have added these metaplots along with APA scores to Supplementary Fig. 1 (please see Suppl Fig 1f). We also amended the main text which now states the following

(Lines 121-123): “On average, the intensity of the E-P, E-E, and P-P interactions (APA score= 2.15, 2.08, and 2.26; Fig. 1i, Supplementary Fig. 1f) was weaker than that of the CTCF loops (APA=2.85; Fig. 1g).”

9-Besides the quantification of enhancer-promoter interactions, a further description of the number of genes affected and their chromosomal location will also help to understand the magnitude and extension of the phenomenon.

- We apologize if this was not clear in the original version of the manuscript but in the **Supplementary Table 3** we provide the list of all genes associated with differential loops and E-P interactions across the oestrous cycle.

10-Reference to (Fig. 3h-i), page 10, line 186, do not match.

- Thank you for noticing this error (it should be Fig. 2h-i instead of Fig. 3h-i). The sentence is now corrected and reads as follows:

(Lines 200-201): “Four of these genes (*Eif2s3x*, *Gjb1*, *Grpr*, and *Tmsb15l*) also showed a higher expression and heightened eigenvector signal in proestrus females compared to males (Fig. 2h-i).”

11-High (i.e., compartments) and middle level (i.e., DNA loops) 3D chromatin changes, besides of lower chromatin changes (i.e., ATAC-seq peaks, Jaric et al., 2019 Nat Commun, PMID: 31253786), do not seem to be aligned. This apparent lack of coherence, and its functional consequences, should be better discussed in the manuscript.

We agree that our results do not fit with a hierarchical model of chromatin organization. Instead, our results agree with recent models where compartments and CTCF loops represent distinct and separate layers of chromatin organization (see our review in Rowley et al. 2018 Nat Rev Genet, see also a preprint concerning this issue using ultra-resolution Hi-C - Gu et al. bioRxiv 2021). This non-hierarchical model is also supported by several studies that have degraded CTCF looping factors with little to no reported change in compartments (Rao et al., 2017 Cell;

Nora et al., 2017 Cell; Wutz et al., 2017 EMBO J; Schwarzer et al., 2017 Nature; Haarhuis et al., 2017 Cell). We have added a discussion of our results in light of these different models.

(Lines 469-484): “Intriguingly, the oestrus cycle- and sex-driven compartment changes on the X chromosome are not composed of altered CTCF loops, and therefore do not support the hierarchical model of chromosome organization. Instead, our results are more consistent with a model where compartments and CTCF loops are independent layers of genome organization¹⁶. This model has been supported by studies where degradation of looping factors results in little to no change in compartments^{30,54}. These earlier studies also showed that de novo identification of chromatin contact domains by Hi-C, sometimes referred to as topologically associating domains (TADs), often represents a mixture of CTCF loop domains and compartment domains^{16,30,55,56}. The definition of “TADs” is of interest, but is discussed elsewhere and is not the purpose of this study⁵⁶. However, we acknowledge that, in recent years, the term “TAD” is becoming more synonymous with CTCF loop domains. Curiously, despite differences in CTCF loops, we do not detect changes in contact domains, i.e. Genova-identified “TADs” nor in CTCF loop domains. This is likely because cycle-specific loops tend to share one or more anchors with unchanging loops. Therefore, the oestrous cycle-driven changes to CTCF loops are likely important for priming the regulation of specific genes nearby, rather than the larger numbers of genes found inside loop domains.”

REVIEWERS' COMMENTS

Reviewer #1 (Remarks to the Author):

The authors have carefully addressed all my concerns and I consider the manuscript ready for publication.

Reviewer #2 (Remarks to the Author):

The authors addressed my concerns adequately.

Reviewer #3 (Remarks to the Author):

The reviewer acknowledges the effort of the authors to investigate and clarify the points raised in the original manuscript. The improved version requires no further modifications.